# Translational fidelity and growth of Arabidopsis require stress-sensitive diphthamide biosynthesis

Hongliang Zhang [1], Julia Quintana [1], Koray Ütkür[2], Lorenz Adrian [3,4], Harmen Hawer[2], Klaus Mayer[5], Xiaodi Gong [6], Leonardo Castanedo[1], Anna Schulten [1], Nadežda Janina [1], Marcus Peters [7], Markus Wirtz [6], Ulrich Brinkmann [5], Raffael Schaffrath [2] & Ute Krämer [1✉]

Diphthamide, a post-translationally modified histidine residue of eukaryotic TRANSLATION ELONGATION FACTOR2 (eEF2), is the human host cell-sensitizing target of diphtheria toxin. Diphthamide biosynthesis depends on the 4Fe-4S-cluster protein Dph1 catalyzing the first committed step, as well as Dph2 to Dph7, in yeast and mammals. Here we show that diphthamide modification of eEF2 is conserved in *Arabidopsis thaliana* and requires AtDPH1. Ribosomal −1 frameshifting-error rates are increased in Arabidopsis *dph1* mutants, similar to yeast and mice. Compared to the wild type, shorter roots and smaller rosettes of *dph1* mutants result from fewer formed cells. TARGET OF RAPAMYCIN (TOR) kinase activity is attenuated, and autophagy is activated, in *dph1* mutants. Under abiotic stress diphthamide-unmodified eEF2 accumulates in wild-type seedlings, most strongly upon heavy metal excess, which is conserved in human cells. In summary, our results suggest that diphthamide contributes to the functionality of the translational machinery monitored by plants to regulate growth.

[1] Molecular Genetics and Physiology of Plants, Faculty of Biology and Biotechnology, Ruhr University Bochum, Universitaetsstrasse 150Box 44 ND3/30, 44801 Bochum, Germany. [2] Microbiology, Institute for Biology, University of Kassel, 34132 Kassel, Germany. [3] Environmental Biotechnology, Helmholtz Centre for Environmental Research—UFZ, 04318 Leipzig, Germany. [4] Chair of Geobiotechnology, Technische Universität Berlin, 13355 Berlin, Germany. [5] Roche Pharma Research & Early Development, Large Molecule Research, Roche Innovation Center Munich, 82377 Penzberg, Germany. [6] Centre for Organismal Studies (COS), University of Heidelberg, 69120 Heidelberg, Germany. [7] Molecular Immunology, Medical Faculty, Ruhr University Bochum, 44801 Bochum, Germany. ✉email: Ute.Kraemer@ruhr-uni-bochum.de

Diphthamide, a post-translationally modified histidine residue, is known in eukaryotic elongation factor 2 (eEF2) proteins of mammals and yeast, and in some archaea[1,2]. The GTPase eEF2 mediates the translocation of the nascent peptidyl-tRNA from the ribosome A site to the P site during protein biosynthesis[3]. Diphthamide was first identified as the target of *Corynebacterium diphtheriae* diphtheria toxin, which catalyzes the ADP-ribosylation of diphthamide, leading to the irreversible inactivation of eEF2, arrest of protein synthesis, and cell death[1,2,4]. Later, *Pseudomonas aeruginosa* exotoxin A, for example, was found to act like diphtheria toxin[5]. In yeast, mice, and human, the evolutionarily conserved pathway of diphthamide biosynthesis requires seven genes, *DPH1* to *DPH7*[1,2]. In the first committed step, the imidazole-$C_2$ atom of a unique histidine residue in eEF2 undergoes the transfer of a 3-amino-3-carboxypropyl (ACP) group from *S*-adenosylmethionine (SAM) through an unconventional radical SAM reaction[6–8] (Fig. 1a). This transfer is mediated by Dph1 in a protein complex with Dph2 and Dph3, also involving Dph4[9–11]. In the second step, Dph5 catalyzes both the trimethylation of the amino group and the methylation of the carboxyl group of the ACP intermediate to produce diphthine methyl ester[12] (Supplementary Fig. 1a). In the final third step of diphthamide biosynthesis, Dph7[12,13] restores the carboxyl group through demethylation to yield diphthine. Subsequently, Dph6[14,15] consumes ATP and an ammonium ion for the amidation of the carboxyl group to generate diphthamide (Supplementary Fig. 1a). The 4Fe-4S cluster-containing Dph1 and Dph2 proteins form a heterodimer[9] and are homologues of a single archaeal Dph2 protein[6]. Crystal structures of archaeal Dph2 of *Pyrococcus horikoshii* and *Candidatus methanoperedens nitroreducens* revealed a homodimer containing one 4Fe-4S cluster per subunit[6,7].

Besides the long-known pathological relevance of diphthamide, the evolutionary conservation and complexity of its biosynthesis suggest a fundamental role of diphthamide in cellular physiology, which remains only partly understood. Insufficient levels or a lack of the diphthamide modification on eEF2 cause severe defects in humans, comprising a developmental delay, short stature, intellectual disability, abnormal skull shape, and altered facial features[16,17]. *Ovarian Cancer 1* ($Ovca1^{-/-} = Dph1^{-/-}$) mutant mice die during embryonic development or at birth, with a developmental delay and defects in multiple organ systems[18]. By contrast, single-celled organisms and mammalian cell cultures that lack diphthamide are viable[19–21]. At the molecular level, elevated rates of −1 ribosomal frameshifting were observed in protein biosynthesis of yeast and mouse mutant cells lacking diphthamide[19,22,23]. Near-atomic resolution cryo-electron microscopy structures of the yeast 80 S ribosome complex containing mRNA, tRNA, and eEF2 provided support for the role of diphthamide in maintaining translational fidelity[24,25].

Here we show that diphthamide modification of eEF2 is conserved in the plant *Arabidopsis thaliana* and requires the function of AtDPH1. Similar to yeast *dph1* mutants, Arabidopsis *dph1* knockouts show enhanced ribosomal −1 frameshifting and hypersensitivity to a protein biosynthesis inhibitor. TARGET OF RAPAMYCEIN (TOR) kinase activity of Arabidopsis *dph1* mutants is attenuated, and autophagy is activated, compared to the wild-type. Different from yeast, Arabidopsis *dph1* mutants are dwarfed as a result of a reduction in cell proliferation in leaves and roots. Finally, abiotic stress, most pronouncedly an excess of the micronutrient copper and the nonessential trace element cadmium, interferes with diphthamide biosynthesis in wild-type plants, and diphthamide modification of human eEF2 is sensitive to copper.

## Results

### The DPH1 protein is conserved in Arabidopsis and required for normal growth

To address the possible existence and biological function of diphthamide in plants, we identified a putative *A. thaliana* DPH1 protein (At5g62030), in which 40% and 28% of amino acid residues are identical to ScDph1 and HsDPH1, respectively (Supplementary Fig. 1a, b). We also identified Arabidopsis homologues of yeast and human Dph2 to Dph7 (Supplementary Fig. 1a). The three cysteine residues thought to be critical for 4Fe-4S cluster binding are conserved in AtDPH1 according to a multiple sequence alignment (C105, C209, and C342; Supplementary Fig. 1b). A neighbor-joining tree reflected the close relationship of putative DPH1 proteins across the plant lineage with human and yeast DPH1, distinct from eukaryotic DPH2 proteins (Supplementary Fig. 1c). Low levels of *AtDPH1* transcript were present in the root, rosette leaf, inflorescence stem, cauline leaf, flowers, and developing silique tissues (Supplementary Fig. 2a), consistent with publicly available gene expression data (http://bar.utoronto.ca/efp2/Arabidopsis/Arabidopsis_eFPBrowser2.html). Histochemical detection of β-glucuronidase (GUS) reporter activity in *AtDPH1*-promoter reporter (pDPH1:GUS) lines suggested *AtDPH1*-promoter activity in all examined tissues, with elevated GUS staining intensity in meristematic regions of the shoot and root tissues (Supplementary Fig. 2b–l). In *dph1-1* and *dph1-2* lines, we confirmed T-DNA insertions in the coding region of the *DPH1* locus by PCR (Supplementary Fig. 2m) and a lack of *DPH1* transcript (Supplementary Fig. 2n), suggesting that *dph1-1* and *dph1-2* are loss-of-function mutants. The aminoglycoside toxin hygromycin B inhibits polypeptide synthesis of bacteria, fungi, and multicellular eukaryotes by binding to peptidyl-tRNA and preventing its EF2-mediated translocation[26]. Yeast *dph2* and *dph4* mutants are hypersensitive to hygromycin treatment[22]. Indeed, yeast mutants lacking any protein involved in the first committed step of diphthamide biosynthesis (*dph1* to *dph4*) were unable to grow in the presence of 80 mg $l^{-1}$ of hygromycin (Supplementary Fig. 2o). The growth of mutants defective in subsequent steps of diphthamide biosynthesis (*dph5* to *dph7*) was slight to moderately reduced at 80 mg $l^{-1}$ and abolished at 120 mg $l^{-1}$ hygromycin, whereas growth of wild-type yeast cells remained almost unaffected. Similar to yeast *dph* mutants, Arabidopsis *dph1-1* and *dph1-2* mutant seedlings were hypersensitive to hygromycin when compared to the wild-type (Fig. 1b). For genetic complementation, we generated homozygous stable *dph1-1* transformant lines with a construct comprising a genomic fragment containing the *DPH1* promoter (1800 bp) and coding region translationally fused to a C-terminal GFP coding sequence (*dph1-1 pDPH1:DPH1-GFP*). *DPH1* transcript levels were similar between the wild-type and four independently transformed *dph1-1* complementation lines (Supplementary Fig. 2p). Complemented line Y22-10 was as tolerant to hygromycin as the wild-type (Fig. 1b). Even on standard media in the absence of hygromycin, fresh biomass and primary root length of Arabidopsis *dph1-1* and *dph1-2* seedlings were ~50% and 40% lower, respectively, than in the wild-type and in three independently transformed complemented lines (Fig. 1c–e). These results show that in Arabidopsis, *DPH1* is required for the maintenance of normal growth rates.

### Cytosol-localized AtDPH1 is necessary for diphthamide modification of eEF2 and contributes to translational fidelity

To examine the subcellular localization of AtDPH1, we imaged roots of *dph1-1 pDPH1:DPH1-GFP* lines by confocal microscopy. Within root tips, the GFP signal was consistent with a cytosolic localization of DPH1, as expected[1,2] (Fig. 2a, Supplementary Fig. 3a). The conserved histidine residue (yeast H699, human H715) that undergoes diphthamide modification corresponds to H700 of *A. thaliana* eEF2 (LOW EXPRESSION OF OSMOTICALLY RESPONSIVE GENES1, LOS1, At1g56070; Supplementary Fig. 3b)[25,27]. Specific antibodies detect either diphthamide-unmodified eEF2 or global eEF2 protein of humans and yeast, in

accordance with identical amino acid sequences in the region targeted by these antibodies[20,22]. We tested whether these antibodies cross-react with Arabidopsis eEF2 in an immunoblot,

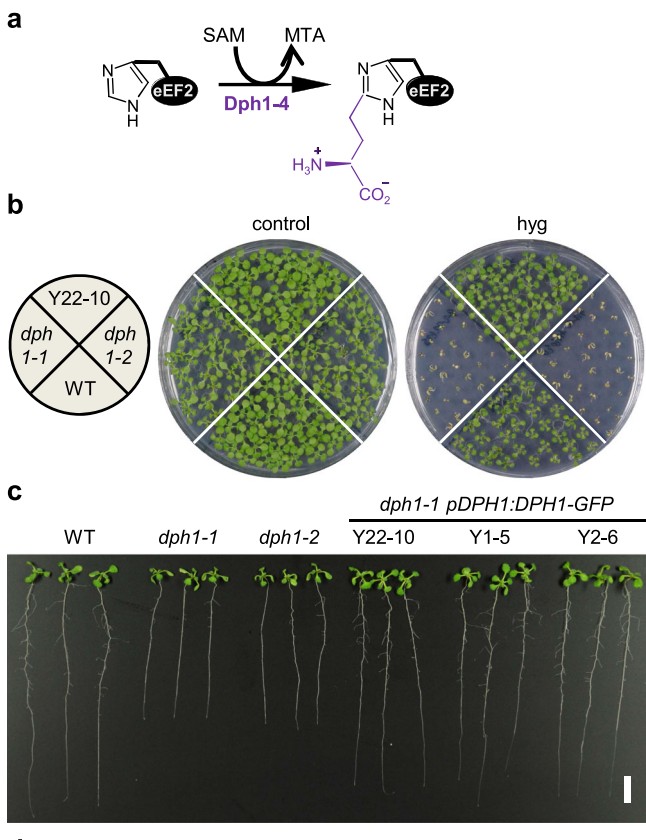

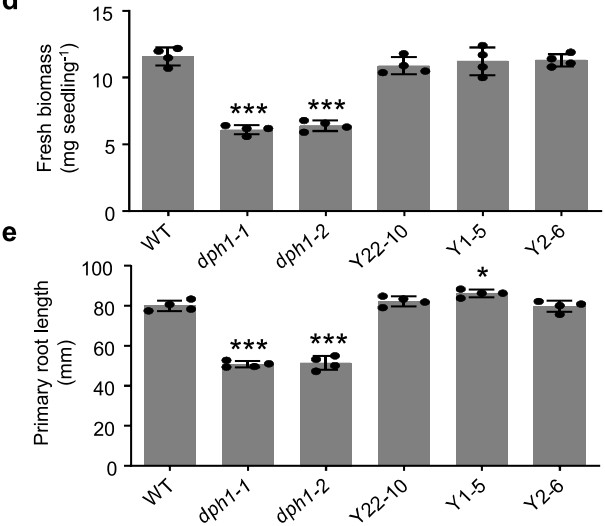

**Fig. 1 Identification and phenotypes of Arabidopsis *dph1* mutants.**
**a** Scheme of first step of diphthamide biosynthesis in yeast and human. SAM S-adenosylmethionine, MTA 5′-methylthioadenosine. **b** Arabidopsis hygromycin sensitivity. Photograph of 12-day-old seedlings of the wild-type (WT), dph1-1, dph1-2, and dph1-1 pDPH1:DPH1-GFP complemented line Y22-10 cultivated without (control) or with 2 μM hygromycin B (hyg). **c–e** Growth of WT, dph1, and complementation lines on 0.5× MS medium, shown as photograph (**c**), fresh biomass (**d**), and primary root length (**e**) of 12-day-old seedlings. Scale bar, 10 mm (**c**). Data are mean ± s.d., $n = 4$ pools of between 5 and 6 seedlings, with each pool sampled from an independent petri plate (**d**, **e**). Significant differences from WT: *$P < 0.05$, ***$P < 0.001$ (one-way ANOVA, Tukey's test).

despite divergent amino acids at positions 1, 4, and 14 of the peptide sequence used to generate the antibodies (Supplementary Fig. 3b). As shown before, unmodified eEF2 can be detected in $DPH1^{-/-}$ in comparison to the wild-type human MCF-7 cell line (Fig. 2b). The same antibody detected unmodified eEF2 in Arabidopsis dph1-1 and dph1-2 mutants, but not in the wild-type or in any of the four complemented lines (Fig. 2b, and see below). In an independent approach, we employed mass spectrometry to identify the post-translational modification of eEF2 at H700. In the wild-type and in the complemented line Y22-10, all informative fragment scans unanimously indicated diphthamide modification of eEF2 (Fig. 2c, Supplementary Fig. 4a). Conversely, we exclusively detected fragment spectra matching unmodified H700 in the dph1-1 and dph1-2 mutants. We estimated that 96% of eEF2 protein carries a diphthamide modification in the wild-type while more than 99% of eEF2 is diphthamide-unmodified in the dph1-1 and dph1-2 mutants, based on a quantitative assessment of the precursor $m/z$ values (±3 ppm) along the chromatographic retention time (Supplementary Table 1). Our data suggested that eEF2 was quantitatively diphthamide-modified in the complemented line Y22-10.

Next, we sought to test for −1 ribosomal frameshifting error in Arabidopsis. We transiently transfected aliquots of wild-type and dph1 mesophyll protoplasts with either one of two dual-luciferase reporter constructs, a control reporter for in-frame translation and a test reporter for programmed −1 frameshifting (Supplementary Fig. 4b–d)[28]. Programmed −1 frameshifting-error rates were elevated by 47% and 39% in dph1-1 and dph1-2 mutants, respectively, compared to the wild-type (Fig. 2d). Global protein biosynthesis rates were not decreased in fully expanded mature leaves of 5-week-old soil-grown dph1 mutant plants (Supplementary Fig. 5). Our results demonstrate that the diphthamide modification of eEF2 is conserved, requires AtDPH1, and contributes to translational fidelity in Arabidopsis.

**Decreased cell proliferation and TARGET OF RAPAMYCIN (TOR) kinase activity in Arabidopsis *dph1* mutants.** Similar to 12-day-old seedlings germinated on agar plates (see Fig. 1c–e), 4-week-old soil-grown dph1 mutants were smaller than wild-type plants (Fig. 3a; rosette diameter of dph1-1 and dph1-2 ~85% and 89%, respectively, of the wild-type), whereby the magnitude of the growth phenotype of dph1 mutants decreased with plant age. The surface area of the sixth oldest fully expanded leaf of dph1 mutants was about 30% smaller than in the wild-type (Fig. 3b, c). There was no significant difference between genotypes in median leaf palisade cell size (Fig. 3d, Supplementary Fig. 6a), whereas the number of cells across the leaf blade was decreased by about one-third in dph1 mutants (Fig. 3e, Supplementary Fig. 6b). Paradermal views and transverse sections of leaves suggested that palisade cells were more irregularly shaped and intercellular spaces enlarged in dph1 mutants compared with the wild-type (Supplementary Fig. 6a,b). Leaf chlorophyll content of dph1 mutants was about 26% lower than in the wild-type (Supplementary Fig. 6c). The number of rosette leaves at 4 weeks of age and at bolting time did not differ between dph1 mutants and the wild-type, suggesting that the rate of formation of new leaves and flowering time were unaffected in the mutants under the employed conditions (Fig. 3f, g). Alterations in endopolyploidy can affect plant size so that we quantified nuclear ploidy levels in leaf cells by flow cytometry. The proportion of 8 C nuclei was significantly increased in dph1 mutants at the expense of 16 C and 32 C nuclei, implying an overall decreased level of endopolyploidy compared to the wild-type (Supplementary Fig. 6d). Primary roots of 9-d-old dph1 seedlings were about one-third shorter than those of the wild-type (Supplementary Fig. 6e, f). Compared with

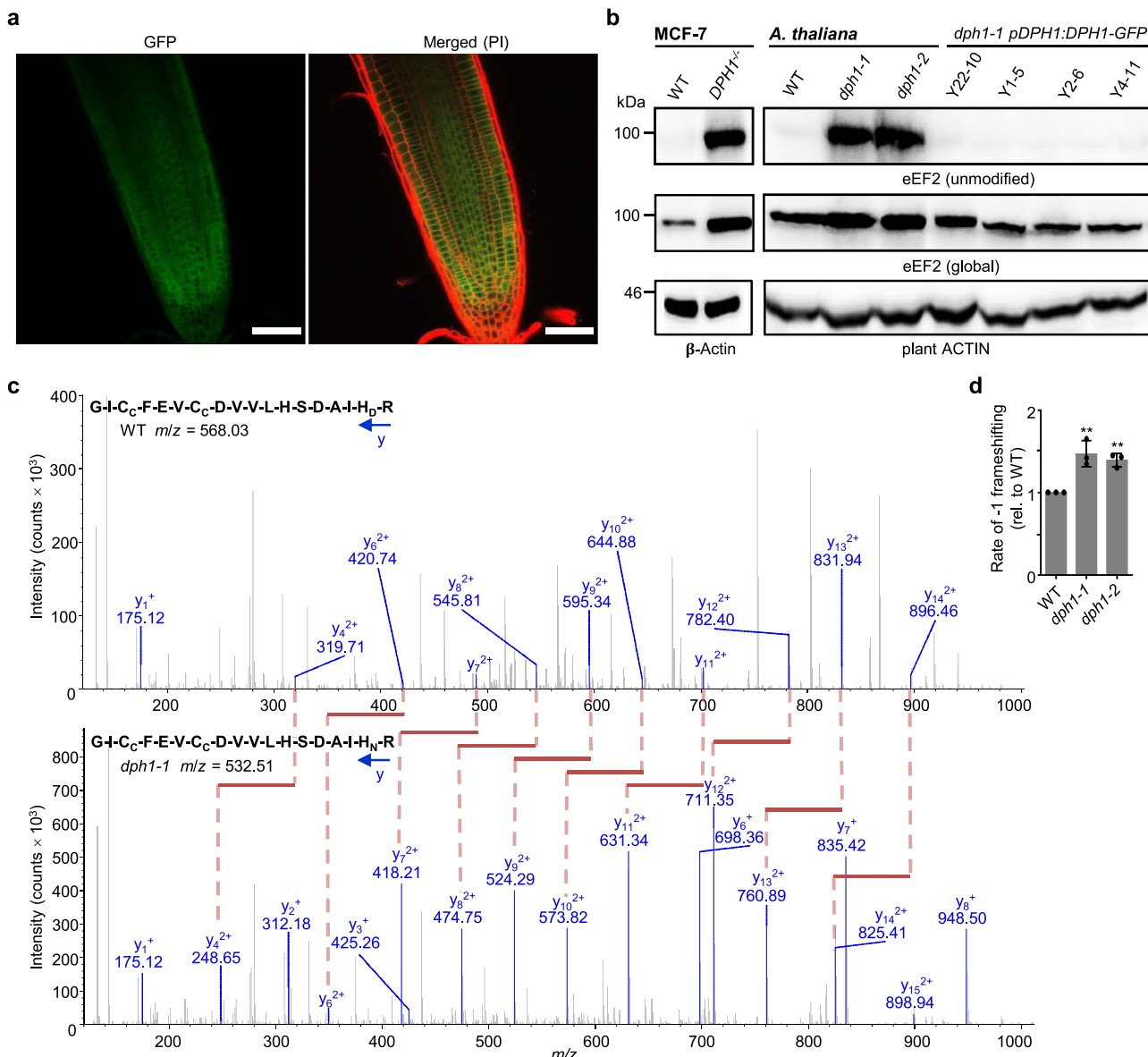

**Fig. 2 Diphthamide modification of eEF2 and translational reading frame accuracy depend on DPH1 that localizes to the cytosol. a** Representative confocal laser scanning microscopic image of a root tip of an 8-day-old *dph1-1 pDPH1:DPH1-GFP* (line Y22-10) seedling stained with propidium iodide (PI, red). Scale bars, 50 μm. **b** Immunoblot probed with antibodies specific for detection of eEF2 lacking the diphthamide modification (unmodified), recognizing all forms of eEF2 (global), or serving as a loading control (anti-β-Actin for MCF-7, anti-plant ACTIN for *A. thaliana*), respectively. Wild-type (WT) and *DPH1⁻/⁻* mutant human cell line MCF-7 served as controls (left). **c** MS/MS spectra of eEF2 peptide 684-GICFEVCDVVLHSDAIHR-701 from WT (top) and *dph1-1* mutant (bottom), with diphthamide (D) or without diphthamide (N) modification on H700 respectively (C: carbamidomethylation). The selected monoisotopic precursor $m/z$ ($z = +4$) is given in each diagram; all detected y fragment ions are shown in blue. Red/pink lines visualize the consistent $m/z$ difference of 71.055 between equivalent $y_n^{2+}$ ions of the two genotypes. Tissues were shoots of 16-day-old 0.5× MS-grown seedlings (**b, c**). **d** Rates of ribosomal −1 frameshifting error. We normalized the ratio of firefly/renilla luciferase activity for a test −1 frameshift reporter construct to the ratio for an in-frame control reporter construct, as measured in transiently transfected Arabidopsis leaf mesophyll protoplasts prepared from 4-week-old soil-grown *dph1* mutants and wild-type plants. Mean ± s.d., $n = 3$ independently transfected replicate aliquots of protoplasts, shown normalized to the wild-type (see Supplementary Fig. 4b-d). Significant differences from WT: **$P < 0.01$ (one-way ANOVA, Tukey's test).

the wild-type, the length of the meristematic zone along the root was about 25% reduced in *dph1* mutants (Fig. 3h, i), with ~40% fewer cortex cells (Fig. 3j) of ~25% increased cell length (Supplementary Fig. 6g). The length of root cortex cells in the differentiation zone was similar between *dph1* mutants and the wild-type (Supplementary Fig. 6h, i).

Following a 30 min application of the modified nucleoside 5-ethynyl-2'-deoxyuridine (EdU), which is incorporated in DNA during replication and detected after harvest upon click coupling to a fluorophore, fluorescence intensity was significantly lower in

root tips of *dph1* mutants, with fewer labeled cells in the root apical meristem, compared to the wild-type (Supplementary Fig. 7a, c). This suggested that the number of cells in S-phase of the cell cycle was reduced in *dph1* mutants. EdU staining was undetectable following treatment with the widely used ATP-competitive specific inhibitor of TOR kinase activity AZD-8055[29], in agreement with an arrest of cell division, as expected[30]. The *pCYCB1;1:GFP-CYCB1;1* cell cycle marker, introduced to homozygosity, suggested that the number of proliferating cells in the G2/M-phase of the cell cycle was significantly decreased in *dph1*

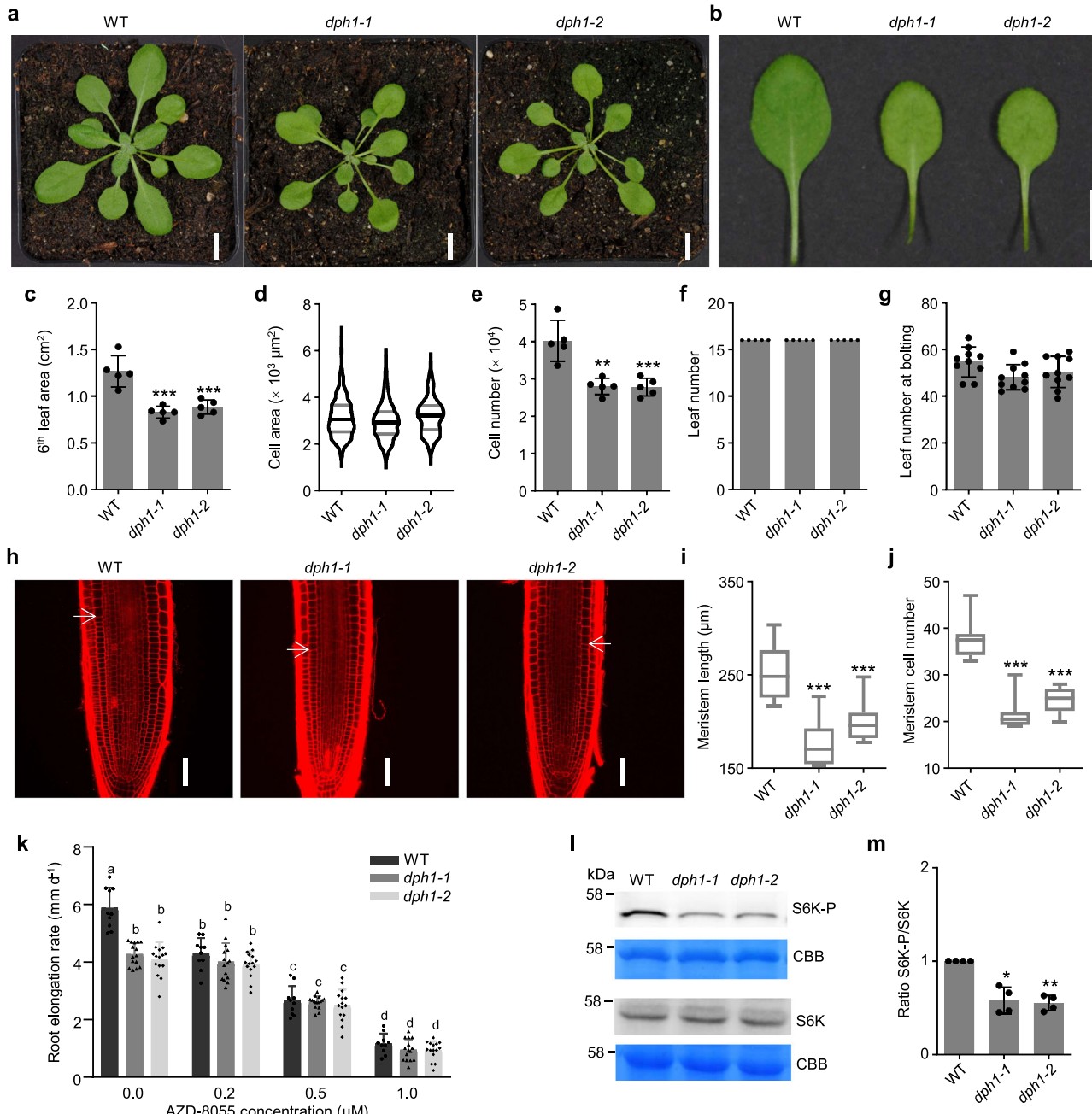

**Fig. 3 Decreased organ size, cell number, and TOR activity in *dph1* mutants. a** Photographs of 4-week-old wild-type (WT) and *dph1* plants cultivated in soil. **b** Sixth oldest leaves of the plants as shown in (**a**). Scale bars, 10 mm (**a**, **b**). **c–e** Leaf area (**c**), cell area (**d**), and cell number (**e**) of the sixth oldest leaves as shown in (**b**). **f** Number of leaves of plants as shown in (**a**). Data are mean ± s.d., *n* = 5 plants (**c,e**, **f**), and a boxplot, *n* = 272 (WT), *n* = 287 (*dph1-1*), and *n* = 255 (*dph1-2*) cells from a total of 5 plants (**d**). **g** Number of leaves at bolting. Data are mean ± s.d., *n* = 10 plants, with cultivation in neutral days (12 h light). **h** Confocal laser scanning microscopic images of PI-stained root meristematic zones of 9-day-old WT and *dph1* mutant seedlings cultivated on 0.5× MS medium. Scale bars, 50 μm. Arrows mark the cortex transition zones. **i, j** Meristem length (**i**) and meristem cell number (**j**) of roots as shown in **h**. Shown are boxplots, *n* = 12 seedlings per genotype. **k** Sensitivity to TOR kinase inhibitor. Data are mean ± s.d., *n* = 10 and 15 seedlings for WT and *dph1* mutants, respectively, from day 4 to day 7 of AZD-8055 treatment (Supplementary Fig. 8). **l, m** TOR activity. Total protein extracts of 14-day-old seedlings grown in liquid 0.5× MS medium with 0.5% (w/v) sucrose were resolved by SDS-PAGE, blotted, and the membrane probed with antibodies against S6K-P and S6K or stained with Coomassie Brilliant Blue (CBB) as a loading control. **m** S6K-P/S6K ratio. Data are mean ± s.d., *n* = 4 independent experiments (**l** and Supplementary Fig. 9). Violin/boxplots show median (center), 1st and 3rd quartile (gray lines/box), minimum and maximum (drawn lines/whiskers) (**d**, **i**, **j**). Significant differences from WT: *$P < 0.05$, **$P < 0.01$, ***$P < 0.001$ (one-way ANOVA with Games-Howell test). Different characters reflect significant differences between means (two-way ANOVA, Scheffé test, $P < 0.05$, **k**).

mutants compared to the wild-type genetic background (Supplementary Fig. 7b, d). Taken together, these data suggest that the smaller size of *dph1* mutants is attributable to decreased cell proliferation in both leaves and roots of juvenile plants, with developmental timing of leaf formation and flowering in older plants remaining similar to the wild-type.

The TOR kinase integrates nutrient availability, energy status, growth regulators and environmental inputs to adjust metabolism, growth, and development across all eukaryotes[30]. Hypersensitivity to TOR kinase inhibitors was reported in yeast deletion strains of several diphthamide biosynthesis genes, alone[31] or in combination with deletion of *EFT2*[22], one of the two gene copies encoding eEF2. In accordance with attenuated TOR activity[32], root elongation rates of *dph1* mutants were tolerant to low levels of 0.2 μM AZD-8055—a concentration that had a significant inhibitory effect on the wild-type (Fig. 3k, evaluated from day 4 to day 7 of exposure, see Supplementary Fig. 8). As a proxy of TOR activity, we assessed the phosphorylation status of the direct TOR target ribosomal protein S6 kinase (S6K)[33]. Immunoblots indicated that the ratios of phosphorylated S6K (S6K-P) to S6K protein amounts were decreased in 14-day-old *dph1* mutant seedlings to about 55% of the ratios in the wild-type (Fig. 3l, m, Supplementary Fig. 9). These results suggest that the lack of diphthamide modification in Arabidopsis *dph1* mutants results in attenuated TOR kinase activity.

**Transcriptome alterations in *dph1* mutants.** To broadly examine the consequences of the lack of diphthamide, we conducted comparative transcriptomics of 4-week-old soil-grown plants using RNA-seq. A principal component analysis separated the transcriptomes of wild-type and *dph1* mutant leaves into distinct clusters (Fig. 4a). Transcripts of 509 genes were differentially abundant in both *dph1* mutants when compared to the wild-type, and of these 267 were upregulated and 242 were downregulated in mutants (Fig. 4b, Supplementary Data 1). Among the transcripts that were more abundant in both *dph1-1* and *dph1-2* than in the wild-type, significantly enriched biological processes were mostly associated with abiotic stress responses, such as "response to cold" and "response to desiccation", whereby "cell wall" and "cytosolic ribosome" were enriched as cellular compartments (Fig. 4c, Supplementary Data 2). Among transcripts less abundant in *dph1* mutants than in the wild-type, biotic stress responses were significantly enriched, for example, "defense response to bacterium" and "systemic acquired resistance", with "plasma membrane" enriched as a cellular compartment (Fig. 4d, Supplementary Data 2). Normalized transcripts per million (TPM) of *DPH1* were 6.52 on average in the wild-type, compared with 0.46 in the two *dph1* mutants, below the threshold 2.46 used to filter out nonexpressed genes (Fig. 4e). The levels of 10 transcripts encoding ribosomal proteins were higher in *dph1* mutant than in wild-type seedlings (Supplementary Data 3). In line with very slightly, but consistently, elevated transcript levels (1.14-fold, TPM WT 570, TPM *dph1* 651, on average; Supplementary Data 1), we observed higher-intensity bands for global eEF2 protein in Arabidopsis *dph1* mutants (see Fig. 2b), as reported earlier in human MCF-7 cells and in yeast[22]. Note that the transcript encoding a putative second eEF2 isoform (At3g12915), was present only at low levels in all genotypes (TPM 4.5 to 4.8; Supplementary Data 1). Eighteen transcripts encoding proteins involved in proteolysis were misregulated in *dph1* mutants, with 11 present at higher levels and 7 present at lower levels than in the wild-type (Supplementary Data 3). As a candidate gene that might contribute to the molecular mechanisms underlying the phenotypic alterations, *TRANSLATIONALLY-CONTROLLED TUMOR PROTEIN 1* (*TCTP1*) transcript levels were reduced by

~520 TPM in *dph1* mutants, with levels of about 57% remaining in comparison to the wild-type (Fig. 4f). In agreement with this, immunoblots suggested 40–50% lower TCTP1 protein levels in *dph1* mutants than in the wild-type (Fig. 4g). Loss-of-function of *TCTP1* is embryo-lethal in Arabidopsis, while *TCTP1* knockdown causes dwarfism and lengthening of the cell cycle with a prolonged G1 phase, and promotes apoptosis[34,35].

**Autophagy is activated, and protein aggregation and ubiquitylation are enhanced, in Arabidopsis *dph1* mutants.** Autophagy is an important, highly conserved process in eukaryotes for the degradation and recycling of proteins, cytoplasmic organelles, and macromolecules to maintain development and growth in response to nutrients and energy deprivation, various stresses, as well as during senescence[36]. Since enhanced −1 frameshifting must lead to the formation of aberrant translational products that are likely to form protein aggregates targeted by autophagy[37] (see Fig. 2d), we tested whether autophagy is activated in Arabidopsis *dph1* mutants. During autophagy, vesicles termed autophagosomes are formed to deliver proteins or organelles to the vacuole for degradation, whereby the number of autophagosomes reflects the level of autophagy[38]. Monodansylcadaverine (MDC) staining of the root elongation zone suggested an enhanced cellular accumulation of autophagosomes in *dph1* mutants compared to the wild-type (Supplementary Fig. 10a, c). Moreover, compared to an AUTOPHAGY-RELATED 8a (ATG8a)-GFP transgenic wild-type line, *dph1-1* ATG8a-GFP seedlings contained a larger number of GFP *punctae* and increased levels of free GFP, as also observed—to an even larger extent—in ATG8a-GFP exposed to the TOR kinase inhibitor AZD-8055 (Supplementary Fig. 10b, d–e). Finally, ATG8 protein lipidation was enhanced by about 30% in *dph1-1* and *dph1-2* compared to the wild-type (AZD-8055 caused an 85% increase), according to immunoblots (Supplementary Fig. 10f). These results support a moderately enhanced autophagic flux in *dph1* mutants compared to the wild-type. NEXT TO BRCA1 GENE1 (NBR1) is a cargo receptor that mediates the selective autophagy of ubiquitylated protein aggregates[39]. According to immunoblots, NBR1 protein levels were slightly, about 1.2-fold higher in *dph1* mutants than in the wild-type (Supplementary Fig. 10g). Note that NBR1 protein levels can increase transiently or remain unchanged, for example upon heat stress[40,41], although they usually decrease when autophagy is induced because NBR1 is an autophagy substrate itself[39]. To test whether the loss of diphthamide triggers autophagy also in yeast, we diagnosed enhanced proteolytic processing of a chimeric GFP-Atg8 fusion protein to free GFP in a *S. cerevisiae dph1* mutant[42], as also observed in wild-type yeast cells treated with the TOR inhibitor rapamycin (Supplementary Fig. 11). Deletion of *ATG1* reversed increased GFP-Atg8 processing both in the *dph1* mutant and under exposure of the wild-type to the TOR inhibitor rapamycin, thus supporting the involvement of autophagy. Taken together, these results confirm that the activation of autophagy is shared by mutants lacking diphthamide of Arabidopsis and yeast.

Compared to wild-type seedlings, *dph1* mutants contained slightly elevated levels of small heat shock proteins HSP17.7 (class I) and HSP17.6 (class II), markers of protein aggregation (Supplementary Fig. 12a, b, f, g)[43]. Misfolded and aggregated proteins are ubiquitylated and can subsequently be processed by the ubiquitin-proteasome system or NBR1-mediated autophagy[41,44,45]. Total levels of diverse protein-ubiquitin conjugates were slightly higher in *dph1* mutant seedlings than in the wild-type (Supplementary Fig. 12c–e, h). Interestingly, *dph1* seedlings were more tolerant to a single pulse of heat stress than the wild-type (Supplementary Fig. 12i). These results detail small but robust alterations in

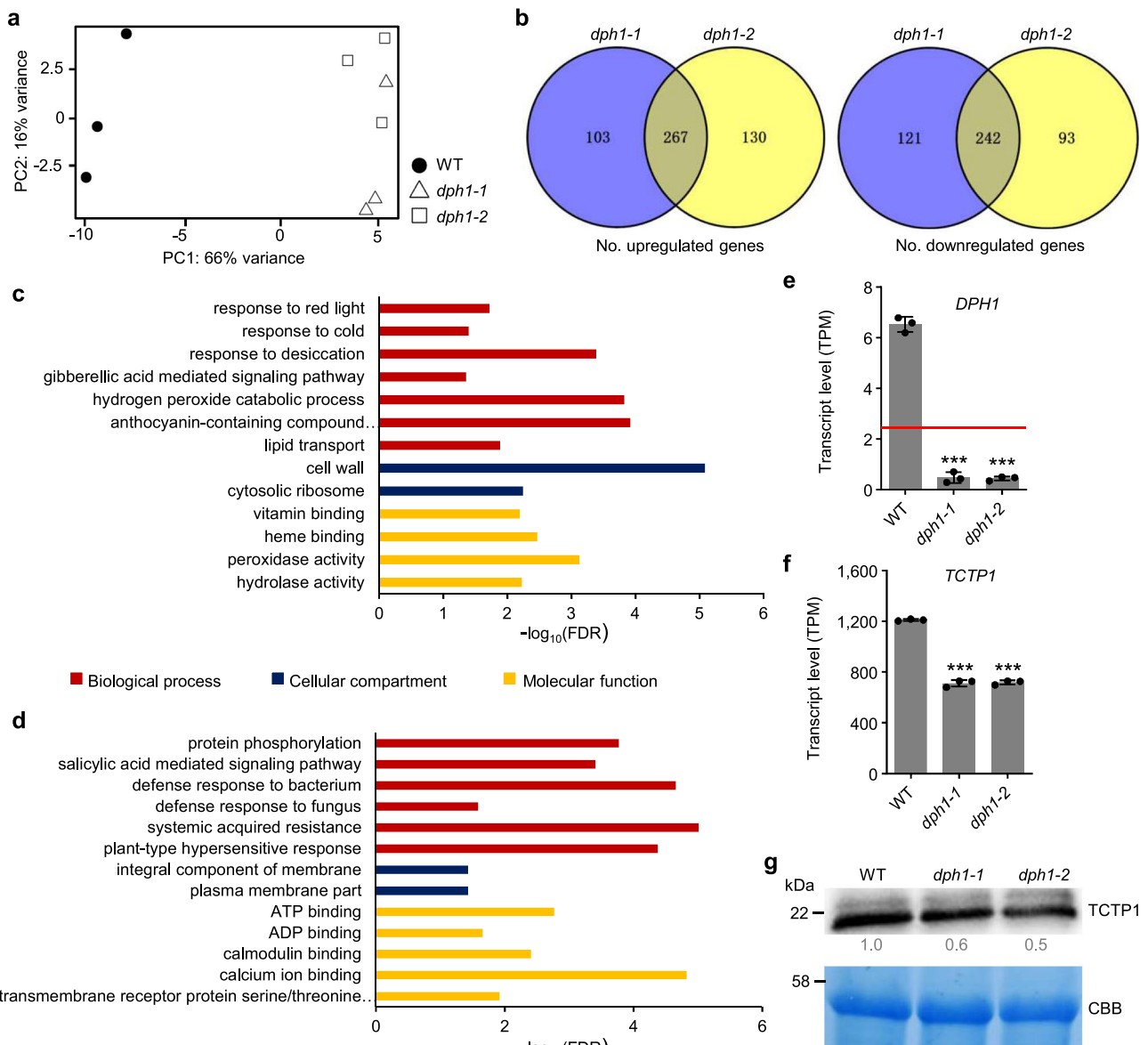

**Fig. 4 Comparative transcriptomics of *dph1* mutants and wild-type plants. a** Principal Component Analysis of normalized transcriptome data (transcripts per kilobase million, TPM). **b** Venn diagrams showing transcripts present at higher (upregulated) and lower (downregulated) levels in *dph1-1* and *dph1-2* mutants, compared to the wild-type (WT). **c, d** Representative GO terms significantly enriched among genes of which transcript levels are upregulated (**c**) and downregulated (**d**) in both *dph1-1* and *dph1-2* mutants compared to WT (FDR < 0.05). **e** Normalized transcript levels of *DPH1*. Red horizontal line marks threshold TPM of 2.46 used to filter out non-expressed genes. **f, g** TPM of *TCTP1* (**f**) and TCTP1 protein levels determined by immunoblot using an anti-TCTP1 antibody or stained with Coomassie Brilliant Blue (CBB) as a loading control; numbers are band intensities relative to those in WT (**g**). Data are mean ± s.d. (**e, f**), $n = 3$ independent experiments, with leaves sampled from 4-week-old soil-grown plants (**a–g**). Significant differences from WT: ***$P < 0.001$ (one-way ANOVA, Tukey's test).

proteostasis of *dph1* mutants that were generally apparent in the transcriptome data (see Supplementary Data 3).

**Exposure to heavy metals interferes with diphthamide modification of eEF2.** Next, we investigated whether environmental conditions can affect diphthamide modification of eEF2 in Arabidopsis. Reactive oxygen species, heavy metal cations with strong ligand-binding affinities, or a lack of iron or sulfur can affect the occupancy of protein Fe-S cluster binding sites in vitro so that these conditions could interfere with DPH1 function and thus with diphthamide modification of eEF2 in planta. The employed set of abiotic stresses decreased shoot fresh biomass by between

19 and 62% of controls (Fig. 5a) and caused partial root growth inhibition or leaf chlorosis (Supplementary Fig. 13a). Immunoblots suggested strongly elevated levels of unmodified eEF2 in wild-type seedlings exposed to copper (Cu) or cadmium (Cd) (Fig. 5b, e–f). Amounts of unmodified eEF2 appeared slightly increased in lead- and oxidative stress-exposed seedlings[46] as well as in iron- and sulfate-deficient mutants[47,48], respectively, when compared to the respective untreated or wild-type controls (Fig. 5b, Supplementary Fig. 13b, c).

Upon cultivation of Arabidopsis on series of Cu and Cd concentrations, we observed progressive decreases in shoot and root fresh biomass as well as primary root lengths (Fig. 5c, d, Supplementary Fig. 14a–c, e–g) and gradual increases in shoot Cu

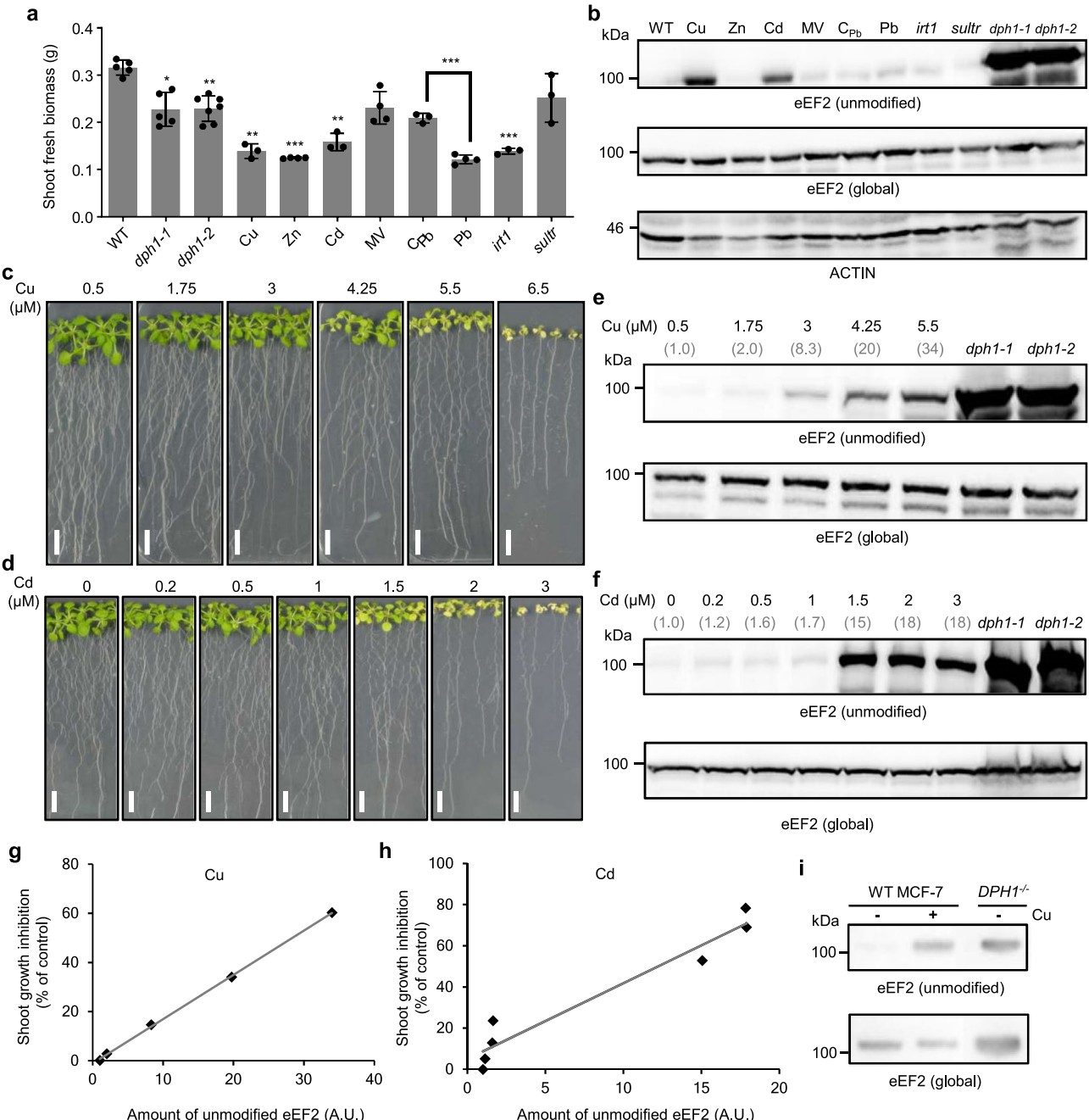

**Fig. 5 Copper and cadmium toxicity correlate with accumulation of diphthamide-unmodified eEF2 protein. a** Shoot fresh biomass of 18-day-old seedlings. Wild-type (WT; $n = 5$), iron uptake-defective *irt1* ($n = 3$), sulfate uptake-defective *sultr1;1 sultr1;2 (sultr)* ($n = 3$), *dph1-1* ($n = 5$), and *dph1-2* ($n = 7$) mutants were cultivated in modified Hoagland medium. Additionally, WT was cultivated in modified Hoagland medium supplemented with 5 µM Cu ($n = 3$), 80 µM Zn ($n = 4$), 2 µM Cd ($n = 3$), 5 nM methyl viologen (MV; $n = 4$), 25 µM Pb ($n = 4$), and a specific Pb-free control medium ($C_{Pb}$; $n = 3$). Shown are mean ± s.d., $n$ biologically independent pools of 17 seedlings, with each pool sampled from a replicate petri plate, from one experiment. *$P < 0.05$, **$P < 0.01$, ***$P < 0.001$ (one-way ANOVA with Games-Howell test, or two-tailed Student's $t$-test compared with the respective control). **b** Immunoblot detection of unmodified and global eEF2 protein in total protein extracts from shoot tissues of the seedlings shown in **a**. **c, d** Photographs of 18-day-old WT seedlings cultivated on series of $CuSO_4$ (**c**) or $CdCl_2$ (**d**) concentrations in modified Hoagland medium (Controls: 0.5 µM Cu, 0 µM Cd). Scale bars, 10 mm. **e, f** Immunoblots (as in **b** for shoots of Cu-exposed (**e**) and Cd-exposed seedlings (**f**), respectively. Numbers indicate the fold increase in unmodified eEF2 protein amount relative to controls, after normalization to global eEF2 protein levels. Protein extracts from *dph1-1* and *dph1-2* seedlings served as controls. **g, h** Relationships between the amount of unmodified eEF2 protein and shoot growth inhibition in Cu-exposed (**g**, $y = 1.80 x - 1.15$, $R^2 = 0.9995$) and Cd-exposed (**h**, $y = 3.68 x + 5.00$, $R^2 = 0.9384$) WT seedlings. A.U. arbitrary units. **i** Immunoblot detection of unmodified and global eEF2 protein in total protein extracts from $DPH1^{-/-}$ mutant and WT of human MCF-7 cell line cultivated in liquid medium without or with addition of 125 µM $CuSO_4$ for 3 days.

or Cd concentrations (Supplementary Fig. 14d, h), paralleled by increasing levels of unmodified eEF2 (Fig. 5e, f). In Cu-exposed seedlings, relative amounts of unmodified eEF2 approximately tripled between 20% and 60% shoot growth inhibition following a linear relationship (Fig. 5g). Cd-exposed seedlings accumulated little unmodified eEF2 up to ~25% shoot growth inhibition, with ~10-fold higher levels of unmodified eEF2 at 60% shoot growth inhibition (Fig. 5h). We observed that diphthamide modification of eEF2 was sensitive to excess Cu in the human MCF-7 cell line (Fig. 5i), but not in yeast (Supplementary Fig. 13d). These results show that not only genetic lesions but also physiological stress can lead to a lack of diphthamide on eEF2 in plants and in human cells.

## Discussion

The diphthamide modification of eEF2 and its biosynthetic pathway are conserved among yeast and mammalian model organisms[1,2], and here we show that they are also conserved in the vascular plant model species *A. thaliana* (Figs. 1 and 2, Supplementary Figs. 1–4). The absence of diphthamide in Arabidopsis *dph1* mutants causes elevated rates of −1 ribosomal frameshifting, and thus plants share the role of diphthamide in translational fidelity that is conserved among fungal and mammalian models[19,22,23]. As shown here, −1 frameshifting increased from ~0.25% in the wild-type to ~0.4% in *dph1* mutants, with rates likely to depend on the sequence of the frameshifting site[28] and possibly also on other endogenous or environmental factors.

The single TOR Ser/Thr kinase multiprotein complex TORC1 of plants is a central regulator of cellular and developmental processes including the cell cycle, translation, autophagy, and growth[30]. The regulation of translation is a well-known output of TOR signaling[30]. Conversely, the attenuated TOR activity in *dph1* mutants (Fig. 3k–m, Supplementary Fig. 9) suggests that the translational machinery can also act directly or indirectly as an input into TOR signaling in plants. Based on the well-established regulatory functions of TOR[30], its decreased activity could account at least in part for the reduced growth of *dph1* mutants (Figs. 1 and 3a–j, Supplementary Fig. 6).

We report here that reduced leaf area and primary root length of Arabidopsis *dph1* mutants result from decreased cell proliferation with fewer cells in the S-phase and in the G2/M-phase of the cell cycle but without global changes in cell sizes (Figs. 1 and 3, Supplementary Figs. 6 and 7). The proportion of endopolyploid cells ≥16 C in *dph1* mutants was lower than in the wild-type. The combination of characteristics observed here in roots of *dph1* mutants is similar to the phenotypes of *retarded root growth* (*rrg*) mutants defective in a gene encoding a mitochondrial protein of unknown function[49].

Interestingly, our data indicated that both *TCTP1* transcript and TCTP1 protein levels are reduced in *dph1* mutants compared to the wild-type (Fig. 4f, g). Decreased proportions of cells showing endopolyploidy ≥16 C, a lengthened cell cycle with a prolonged G1 phase, decreased cell proliferation with intact cell elongation, and dwarfing, were reported in *TCTP1* RNAi lines[34], very similar to *dph1* mutants. *TCTP* RNAi in *Drosophila melanogaster* caused decreased cell size, cell number, and organ size, as well as additionally reduced levels of S6K phosphorylation[50]. Drosophila *TCTP* was able to complement an Arabidopsis *tctp1* mutant, and vice versa[34], implying remarkable functional conservation across kingdoms. TORC1 of animals is activated by the Ras superfamily GTPase Rheb (Ras homologue enriched in brain). Rheb was proposed to be positively regulated by TCTP acting as Guanine Exchange Factor (GEF) in Drosophila and human, yet apparently with a strong but poorly understood dependence on the experimental conditions[50–53]. Since a Rheb

orthologue is missing in Arabidopsis, researchers hypothesized that another small GTPase acts as the functional equivalent of Rheb[54] and that there could be a differing pathway for TCTP1 to regulate TOR activity[35]. To date, the evidence for a TCTP1-dependence of TOR activity in plants is merely indirect[35]. Decreased TOR activity can suppress translation of the *TCTP1* transcript, but not its abundance[55], different from our observation of decreased *TCTP1* transcript levels in *dph1* (Fig. 4f, g).

The activation of autophagy in *dph1* mutants (Supplementary Fig. 10) could be attributed to decreased TOR activity[56], or alternatively or additionally to an accumulation of misfolded or aggregated peptides in *dph1* mutants, conceivably as a result of translational frameshifting (Figs. 2d and 3l, m, Supplementary Figs. 9, 10, 12). Our transcriptomics accessed long-term effects in the steady-state (Fig. 4). The lack of diphthamide acts indirectly to cause a number of alterations including moderate changes in TOR kinase activity and autophagic flux. We would not expect these changes to necessarily affect all of the previously reported downstream outputs of TOR signaling in concert, especially given also the respective differing response and developmental dynamics, environmental conditions, and analyzed tissues[55–57]. Future work should address how our observations on Arabidopsis *dph1* mutants are causally linked, for example by addressing whether reduced TCTP1 levels lead to the attenuation of TOR activity, as the literature may suggest, and whether decreased TOR activity contributes to the cell cycle and growth phenotypes of *dph1* mutants.

Physiological stress is apparent in *dph1* mutants (Fig. 4), and this could be an alternative possible cause of decreased TOR activity, as is known in Arabidopsis[30]. Heat stress causes protein misfolding, which promotes the accumulation of the affected proteins in cytotoxic protein aggregates. Subsequently, the dynamic coordination between protein disaggregation and proteolysis enables plants to survive[41,44,45]. The preactivation of some of these processes might give *dph1* mutants an advantage that results in enhanced tolerance to heat stress and possibly also to other disruptions of proteostasis (Supplementary Fig. 12)[43].

Among a variety of nutrient deficiency, oxidative, and heavy metal abiotic stresses, we found that Cu and Cd exposure resulted in a marked concentration-dependent accumulation of unmodified eEF2 in wild-type Arabidopsis seedlings, which correlated with toxicity symptoms (Fig. 5, Supplementary Figs. 13, 14). The Dph1-Dph2 heterodimer that catalyzes the first step in diphthamide biosynthesis is thought to require per monomer one 4Fe-4S cofactor located at the protein surface[6,7]. In bacteria, accessible Fe-S clusters of proteins can be sensitive to displacement by heavy metal cations[58], but the primary molecular mechanisms of heavy metal toxicity in eukaryotes remain largely unknown. Because the diphthamide modification of eEF2 is thought to be irreversible[19] and is a product of a linear biochemical pathway, diphthamide is well suited as a reporter of disruption of its own biosynthesis in vivo. The identification of diphthamide biosynthesis as a target of abiotic stress in Arabidopsis, most pronouncedly from exposure to an excess specifically of the heavy metals Cu and Cd, may have implications for human health (Fig. 5i, further information in Supplementary Figs. 15 and 16)[18].

In summary, we show that diphthamide modification of eEF2 is conserved in plants, together with its role in the maintenance of translational fidelity and the requirement of *DPH1* for diphthamide biosynthesis. Moreover, Arabidopsis requires *DPH1* function to maintain cell proliferation at levels sustaining normal growth rates, as well as the full capacity for TOR activation. Environmental stress, most pronouncedly exposure to an excess of the heavy metals Cu and Cd, causes decreased diphthamide modification of eEF2 in wild-type Arabidopsis. Our work

provides insights into the coordination between translation and growth in plants, and it identifies diphthamide biosynthesis as a molecular target of environmental stress in Arabidopsis and human cells.

## Methods

**Plant material and growth conditions**. *Arabidopsis* Columbia-0 (Col-0, wild-type), *dph1-1* (SALK_205272C), and *dph1-2* (SALK_030334) in the Col-0 background were from NASC (http://arabidopsis.info/). The *irt1* mutant (*pam42*)[48] and the *sultr1;1 sultr1;2* double mutant[47] were previously characterized. Seeds were surface-sterilized in chlorine gas[59] for 4 h and sown on 120 mm square polystyrene petri dishes (Greiner Bio-One, Frickenhausen, DE) containing either 0.5× MS salts with vitamins (Murashige and Skoog, M0255, Duchefa, Haarlem, NL), or modified Hoagland solution[60] supplemented with 5 µM ZnSO$_4$ (instead of 1 µM), each with 1% (w/v) sucrose and 0.8% (w/v) agar (Type M, Sigma, Steinheim, DE), followed by cultivation with the vertical orientation of petri plates. For TOR kinase inhibitor AZD-8055 sensitivity assays, seedlings of WT and *dph1* mutants were cultivated in 0.5× MS medium for 1 week, followed by transfer to fresh medium supplemented with 0, 0.2, 0.5, or 1 µM AZD-8055 (Hycultec, Beutelsbach, DE). For hygromycin sensitivity assays, sterilized seeds were sown on agar-solidified 0.5× MS medium (see above) in round 90 mm polystyrene petri dishes (Greiner Bio-One) with or without an additional 2 µM hygromycin B (Duchefa), followed by cultivation with the horizontal orientation of petri plates. Plates were kept at 4 °C in darkness for 2 days before transfer into a growth chamber and cultivation in 12 h light (120 µmol photons m$^{-2}$ s$^{-1}$, 22 °C)/12 h dark (18 °C; CLF Plant Climatics, Emersacker, Germany) cycles. For CuSO$_4$ (Merck, Darmstadt, DE), ZnSO$_4$ (Applichem, Darmstadt, DE), CdCl$_2$ (Sigma), and methyl viologen (Sigma) treatments, sterilized seeds were germinated and cultivated on agar-solidified Hoagland medium (see above), with additions as indicated. The herbicide methyl viologen (paraquat) causes light-dependent oxidative stress in plants[46]. Pb(NO$_3$)$_2$ (Sigma) treatments were done in an altered modified Hoagland medium at a lower pH (0.3 mM acetic acid-sodium acetate buffer, pH 4, instead of 3 mM 2-(N-morpholino ethanesulfonate, MES, buffer, pH 5.7) and with only 0.093 mM KH$_2$PO$_4$[61]. For heat tolerance tests, one-week-old seedlings of wild-type, *dph1* mutants, and *dph1* complementation line (Y22-10) grown in agar-solidified 0.5× MS medium on round petri plates (see above) were placed into an incubator at 50 °C in the dark for 50 min and subsequently returned to normal growth conditions for 3 days. For cultivation in soil, seeds were first germinated in agar-solidified 0.5× MS medium (see above) for 2 weeks before transfer into the soil (Minitray, Balster Einheitserdewerk, Fröndenberg, DE) and cultivation in a growth chamber in 12 h days (temperatures and light intensity as above). Deviating from this, plants were cultivated for 11 h days before protoplast isolation and for long days (16 h) for RT-PCR and part of the GUS staining experiments. Unless stated otherwise, plant tissues were harvested, immediately frozen in liquid nitrogen, and subsequently stored at −80 °C. Before use, tissues were ground to a powder with a pestle and mortar without thawing. All PCR primer sequences are listed in Supplementary Table 2.

**RNA isolation, cDNA synthesis, RT-PCR, and RT-qPCR**. Total mRNA was extracted using TRIzol (Fisher Scientific, Schwerte, DE), except for RNA isolation from siliques by RNeasy Plant Mini Kit (Qiagen, Hilden, DE). Reverse transcription was carried out on 2 µg DNase I-digested (New England Biolabs, Frankfurt, DE) total RNA using the SuperScript III kit (Fisher Scientific) with oligo(dT) primer following the manufacturer's instructions. For RT-PCR, cDNAs were used as templates for PCR using DreamTaq Green DNA-Polymerase (Fisher Scientific). RT-qPCR was performed using GoTaq qPCR Master Mix (Promega, Walldorf, DE) on a LightCycler480 (Roche, Mannheim, DE).

**Transgenic plants**. For genetic complementation of the Arabidopsis *dph1-1* mutant and DPH1 subcellular localization, the construct *pDPH1:DPH1-GFP* was generated through GreenGate cloning[62]. Briefly, the promoter region (1.8 kb upstream of the translational start codon), the coding sequence, and the 3′ UTR sequence (548 bp downstream of the stop codon) of *DPH1* were separately cloned into the appropriate pUC19-based entry vectors via *BsaI* sites. Subsequently, the three completed entry vectors, together with pGGB003 (*B-dummy*), pGGD001 (*linker-GFP*), and pGGF008 (*pNOS:BastaR:tNOS*), were assembled into a pGreen-IIS based destination vector PGGZ003 via the *BsaI* sites to obtain a translational GFP fusion construct of *DPH1* under the control of the native promoter. *Agrobacterium tumefaciens* (GV3101) mediated transformation of *dph1-1* plants was conducted as described[63]. To generate *pDPH1:GUS* plants, the 1.8 kb promoter region and the 548 bp 3′ UTR region of *dph1* were cloned into entry vectors. The two completed entry vectors, together with pGGB003 (*B-dummy*), pGGC051 (*GUS*), pGGD002 (*D-dummy*), and pGGF008 (*pNOS:BastaR:tNOS*), were assembled into the pGreen-IIS based destination vector PGGZ003 to generate the *pDPH1:GUS* construct employing GreenGate cloning[62], followed by *Agrobacterium tumefaciens*-mediated stable transformation of Col-0[63].

**Immunoblots**. Tissue powder was homogenized in 2× volume of Laemmli buffer (125 mM TRIS-HCl (pH 6.8), 10% (v/v) mercaptoethanol, 0.01% (v/v) bromophenol blue, 4% (w/v) SDS, and 20% (v/v) glycerol) and kept at 100 °C for 10 minutes. After centrifugation, supernatants were separated by 7.5–15% (w/v) SDS-PAGE and blotted onto a polyvinylidene difluoride (PVDF) membrane by wet/tank transfer[59]. For the detection of global and diphthamide-unmodified eEF2[20,22], the PVDF membrane was blocked with 5% (w/v) BSA (bovine serum albumin) in TBS-T (0.05% Tween-20). The blocked membrane was probed with global eEF2 antibody (3C2, 1:4800, Roche) or diphthamide-unmodified eEF2 antibody (10G8, 1:4800, Roche), followed by incubation with the secondary antibody goat anti-rabbit IgG/HRP (1:10,000, 12-348, Fisher Scientific). For the detection of Arabidopsis ACTIN, NBR1, HSP17.7, HSP17.6, TCTP1, GFP, and ubiquitin-conjugated proteins, the PVDF membrane was blocked with 5% (w/v) low-fat milk powder (AppliChem) in TBS-T (0.1% Tween-20). The blocked membrane was probed with anti-ACTIN (1:5000, AS13 2640, Agrisera, Vännäs, SE), anti-NBR1 (1:5000, AS14 2805, Agrisera), anti-HSP17.7 (1:1000, AS07 255, Agrisera), anti-HSP17.6 (1:1000, AS07 254, Agrisera), anti-UBQ11 (1:2000, AS08 307, Agrisera), anti-GFP (1:5000, AB10145, Sigma), or anti-TCTP1[34] (1:5000), followed by incubation in the secondary antibody goat anti-rabbit IgG/HRP (Fisher Scientific). For human Actin detection, the PVDF membrane was blocked with 5% (w/v) BSA in TBS-T (0.05% Tween-20). The blotted membrane was probed with anti-β-Actin (1:5000, A 5441, Fisher Scientific) followed by secondary antibody goat anti-mouse IgG/HRP (1:5000, 12-349, Fisher Scientific) incubation. Immunological detection of S6K-p and S6K was performed as described previously[33], with minor modifications. In brief, total soluble proteins were extracted from 50 mg of ground frozen seedlings with 250 µl Laemmli buffer supplemented with 1% (w/v) phosphatase inhibitor cocktail 2 (abcam, Berlin, DE) and 1x protease inhibitor cocktail (Roche). Proteins were denatured at 90 °C for 10 min and separated on 10% (w/v) SDS-PAGE followed by wet/tank transfer to PVDF membranes. Membranes were blocked with 5% (w/v) BSA (S6K-P) or 7.5% (w/v) low-fat milk powder (S6K) in TBS-T (0.05% Tween-20). Membranes were incubated with primary antibody anti-S6K1 (phospho T449) in 1% BSA (1:5000, ab207399, abcam) or anti-S6K in 5% (w/v) low-fat milk powder (1:5000, AS12-1855, Agrisera) at 4 °C overnight, and subsequently with secondary antibody goat anti-rabbit IgG/HRP (1:5000, 12-348, Fisher Scientific) in 1% (w/v) BSA or 5% low-fat milk powder (as during primary antibody incubation), respectively. For ATG8 and ATG8-PE detection, proteins were separated by 15% (w/v) SDS-PAGE with 6 M urea in the resolving gel. Anti-ATG8 was used as the primary antibody (1:5000, AS14 2811, Agrisera), followed by procedures as described for Arabidopsis ACTIN. All membranes were blocked at the temperature used for antibody incubations for 1 h, and all antibodies were added to a solution of the same composition as used for blocking followed by incubation at RT for 1 h, with any exceptions stated above. Triple 10-min washes were conducted in TBS-T alone composed during blocking after each antibody incubation. All signals were detected with ECL select western blotting detection reagent (GE Healthcare, Little Chalfont, UK) and a Fusion Fx7 GelDoc (Vilber Lourmat, Eberhardzell, DE). Band intensities were quantified using ImageJ[64].

**Detection of the diphthamide modification in eEF2 by mass spectrometry**. Total protein was extracted from shoot tissues and separated by SDS-PAGE (7.5% w/v polyacrylamide) as described above, stained with Coomassie Brilliant Blue, and a gel band corresponding to the size of eEF2 (90–105 kDa) was excised from the gel. Disulfides were reduced with dithionite and cysteine residues were alkylated with iodoacetamide, followed by trypsin digestion of proteins overnight, all within the gel piece as described[65]. Trypsin-digested fragments were eluted from the gel pieces and desalted using ZipTips[66] before analysis by nano-liquid chromatography tandem mass spectrometry (nLC-MS/MS) on a Thermo Orbitrap Fusion mass spectrometer (ThermoFisher) with injection via an electrospray ion source (Tri-Versa NanoMate, Advion). In the Orbitrap detector, MS1 scans were acquired at a resolution of 120,000, with a scan range from m/z 350 to 2000, maximum injection time of 50 ms, automatic gain control (AGC) target of 4 × 10$^5$ ions, dynamic exclusion of 30 s and a minimum intensity 5 × 10$^4$ ions for precursor picking. MS2 (fragment) scans were recorded in the Orbitrap detector at a resolution of 60,000, isolating precursors in a 1.6-Da window, fragmentation at 30% energy High Energy Collision Dissociation (HCD) and 5% stepped collision energy, with 120 ms injection time maximum and an AGC target of 5 × 10$^4$ ions. Diphthamide-modified and diphthamide-unmodified precursor masses of 2269.122 and 2127.011, respectively, of the target peptide 684-GICFEVCDVVLHSDAIHR-701 were identified with ProteomeDiscoverer Version 2.4 (ThermoFisher) using SequestHT as the search engine and the sequence of eEF2 (At1g56070) of *Arabidopsis thaliana* as database. Parameters included carbamidomethylation of cysteine as a fixed and the diphthamide modification of histidine (+C$_7$H$_{14}$N$_2$O, m = 142.11 g) as a variable modification. We allowed no missed cleavage, a precursor charge state of +2 to +7, a precursor m/z tolerance of ±3 ppm, and a fragment mass tolerance of 0.1 Da. The false discovery rate was set to 1% at the peptide identification level using the Target Decoy PSM Validator node. Precursor abundance was estimated with the Minora node in ProteomeExplorer.

**DPH1 subcellular localization**. Roots of 8-day-old seedlings of *dph1-1* *pDPH1:DPH1-GFP* complementation lines were stained with 0.01 g l$^{-1}$ propidium

iodide (Sigma) for 1 min, then briefly washed with ultrapure water. GFP florescence was detected using a Leica SP5 confocal laser scanning microscope, with an excitation wavelength 488 nm, and detection using GFP settings (Leica, Wetzlar, DE). Three T2 seedlings were imaged for each of the five independent complementation lines, alongside Col-0 as a negative control.

**Morphological and cellular analyses.** For palisade cell size measurement, leaves were first cleared in a mixture of 10 ml glycerol, 80 g chloral hydrate, and 30 ml $H_2O$, and photographed using a microscope in differential interference contrast mode (Imager.M2, Zeiss, Jena, DE). Resin embedding and transverse sectioning of resin-embedded leaves were conducted as described[67]. The transverse sections were stained with 0.05% (w/v) toluidine blue for 10 min and rinsed with distilled water for 1 min. Microscopic images were taken in a bright field (Imager.M2, Zeiss). For determining the meristematic region, the roots of 9-day-old seedlings were dipped into 10 µg ml$^{-1}$ propidium iodide for 1 min and briefly rinsed in ultrapure water. Roots were imaged with a Leica SP5 confocal laser scanning microscope (Leica) using a ×20 objective, with excitation through the 488 nm laser and detection using propidium iodide settings (Leica). Leaf areas, meristem length, cell sizes, and cell numbers were determined through ImageJ[64] on photographic images.

**Frameshifting assays in protoplasts.** A dual-luciferase reporter system developed in yeast with a control reporter (pJD375) and a LA-Virus site-based −1 frameshift reporter (pJD376)[28] was adapted for transient expression assays in Arabidopsis mesophyll protoplasts. The *renilla* and *firefly luciferase* cDNAs, together with either the polylinker region from pJD375 or the programmed frameshifting signal from pJD376, were amplified with primers introducing *XhoI* and *SpeI* sites. The amplified fragments were cloned into the *XhoI* and *SpeI* sites of transient expression vector pMatrix[68]. The resulting plasmids were used to transfect mesophyll protoplasts isolated from soil-grown wild-type or *dph1* mutant plants as described[69]. Transfected protoplasts were harvested by centrifugation after dark incubation at 22 °C for 16 h. Protoplasts were disrupted through two freeze-thaw cycles. Firefly and renilla luciferase activities were quantified using the dual-luciferase reporter assay system (Promega) in a Synergy HTX microplate reader (BioTek, Bad Friedrichshall, DE).

**Transcriptome sequencing.** Rosette tissues of 4-week-old plants grown in soil were harvested for total RNA isolation by RNeasy Plant Mini Kit (Qiagen). Libraries were prepared and sequenced using the Illumina platform by Novogene (Hongkong, CN). For RNA-seq data analysis, raw reads were first filtered by CutAdapt 2.1[70] to remove adaptors and low quality reads. The filtered reads were mapped to *Arabidopsis thaliana* Col-0 reference genome (Tair 10). Read counts were determined by Qualimap2[71]. Differentially expressed genes were identified using the R package *DESeq2*[72]. AgriGO (http://systemsbiology.cau.edu.cn/agriGOv2/index.php) was used for gene ontology (GO) term enrichment analyses with the Yekutieli (FDR under dependency) multitest adjustment method[73].

**Sequence alignment and phylogenetic analysis.** Dph1 to Dph7 proteins of *S. cerevisiae* served as query sequences to retrieve protein homologs from the NCBI protein database (https://www.ncbi.nlm.nih.gov/) using blastp[74]. Multiple sequence alignment and the following construction of a neighbor-joining tree of DPH1 homologs were performed with MEGA6.0 software[75]. Bootstrap analysis was performed with 1000 iterations.

**Detection of GUS activity by histochemical staining.** Freshly harvested tissues of homozygous *pDPH1:GUS* lines (T3 generation) were immersed in GUS staining buffer (0.2% (v/v) Triton X-100, 50 mM sodium phosphate buffer pH 7.2, 2 mM potassium ferrocyanide, 2 mM cyclohexylammonium 5-bromo-4-chloro-3-indolyl-β-D-glucuronate, X-Gluc) at 37 °C for 24 h after vacuum infiltration for 1 min. The samples were then immersed in 75% (v/v) ethanol overnight to remove chlorophyll before photographing using a microscope (VS120, Olympus, Hamburg, DE).

**Yeast strains, assays, and cultivation conditions.** Yeast strains used in this study were described before[22]. For growth assays, cell suspensions (OD$_{600}$: 1.5) were tenfold serially diluted and spotted onto YPD medium containing hygromycin B (0, 80, or 120 mg l$^{-1}$) and incubated at 30 °C for 2 days. The GFP-Atg8 cleavage assay diagnostic for autophagy in yeast was performed as described[42]. To test the effect of Cu on diphthamide modification, yeast strains cultivated at 30 °C were harvested at an OD$_{600}$ of 1. Mammalian MCF-7 cells were cultivated in RPMI/10% FCS at 37 °C in humidified 5% $CO_2$[20,76] without or with addition of 125 µM CuSO$_4$ for 3 days.

**Quantification of global translation rates.** This was done as described, including prior plant cultivation in short days[77]. In brief, one leaf disc (diameter 7 mm) was excised from each of the four largest rosette leaves per plant, with three replicate plants per genotype, of 5-week-old soil-grown plants. Leaf discs were floated on incubation medium (2.5 mM Ca(NO$_3$)$_2$, 0.5 mM MgSO$_4$, 2.5 mM KNO$_3$, 0.5 mM KH$_2$PO$_4$, 4 µM FeEDTA, 25 µM H$_3$BO$_3$, 2.25 µM MnCl$_2$, 1.9 µM ZnCl$_2$, 0.15 µM CuCl$_2$, 50 nM (NH$_4$)$_6$Mo$_7$O$_{24}$, pH 5.8) supplemented with 70 µCi ml$^{-1}$ EasyTag™ EXPRESS$^{35}$S protein labeling mix (43.48 Bq fmol$^{-1}$, PerkinElmer, Rodgau,

Germany) in the light (100 µE, 20 °C) for up to 90 min, beginning at ZT 4 h. Leaf disks were harvested at the indicated time points, washed in an incubation medium, and immediately snap-frozen in liquid nitrogen. Frozen leaf tissues were ground to a fine powder, followed by the addition of 0.3 ml extraction buffer (50 mM HEPES, 10 mM KCl, 1 mM EDTA, 1 mM EGTA, 10% (v/v) glycerol, pH 7.4) and extraction on ice for 15 min, with vortexing every 2 min. Suspensions were centrifuged (15,000 × *g*, 4 °C, 15 min). A volume of 0.15 ml of the supernatant was mixed with an equal volume of extraction buffer, followed by the removal of excess Easy Tag™ label and low-molecular-mass metabolites using a PD Spintrap™ G-25 column (Cytiva, Freiburg, DE). An aliquot (50 µl) of the protein solution was added to 10 ml liquid scintillation cocktail (Ultima Gold™, PerkinElmer), and the protein-incorporated isotope label was quantified using a Tri-Carb 2810TR Liquid Scintillation Analyser (PerkinElmer).

**Chlorophyll content index.** Chlorophyll content index is described as the ratio of the proportional transmittance at the wavelengths of 931 nm and 653 nm from chlorophyll absorbances in leaf tissue, and was quantified in the sixth oldest leaf of 4-week-old plants using a Chlorophyll Content Meter with default settings (CCM-200 plus, Opti-Sciences, Hudson, NH, USA).

**Flow cytometry.** Nuclei were isolated as previously described with minor changes[78]. Briefly, the first pairs of true leaves of 21-day-old soil-grown plants were chopped with a razor blade in precooled buffer (10 mM MgSO$_4$, 50 mM KCl, 5 mM HEPES, 10 mM DTT, 2.5% (v/v) Triton X-100) to release nuclei, and the homogenates passed through a 30 µm CellTrics filter (Sysmex Partec, Münster, DE) to remove debris. Nuclear DNA was stained in 0.1 mg ml$^{-1}$ propidium iodide while RNA was removed by 0.1 mg ml$^{-1}$ RNase A (Fisher Scientific). The stained nuclei were run on a CyFlow SL 3-Colour FCM System (Sysmec Partec). Twenty thousand nuclei were analyzed per sample (for gating strategy see Supplementary Fig. 17).

**Sensitivity analyses of wild-type (WT) and *dph1* mutants to AZD-8055.** Images were taken 4 and 7 days after transfer onto AZD-8055-containing medium with a camera (NIKON D200). Primary root lengths were quantified using ImageJ[64].

**EdU staining.** Ten to twelve 8-day-old seedlings cultivated in 0.5× MS medium were transferred per well of polystyrene six-well plates containing 2 ml liquid 0.5× MS salts with vitamins (Murashige and Skoog, M0255, Duchefa, Haarlem, NL) supplemented either with 20 mM Glucose or alternatively with 5 µM AZD-8055, and transferred back into the growth chamber for 90 min. Subsequently, EdU (5-ethynyl-2′-deoxyuridine) staining was initiated using Click-iT® EdU AlexaFluor® 488 Imaging Kit following the manufacturer's protocol at ZT 6 h (Invitrogen, Waltham, MA USA). Accordingly, seedlings were transferred onto agar-solidified 0.5× MS medium in 120 mm square polystyrene petri dishes, and 5 µl of a 1 µM EdU stock solution freshly prepared in liquid 0.5× MS salts with vitamins containing 0.5% (w/v) sucrose was added directly onto the root tip of each seedling, followed by incubation in the growth chamber (120 µmol photons m$^{-2}$ s$^{-1}$, 22 °C) for 30 min. Subsequently, seedlings were fixed in 100 µl fixation/permeabilization solution, consisting of 4% (v/v) formaldehyde and 0.1% (v/v) Triton X-100 in PBS, for 30 min, followed by washing in PBS (3 times 2 ml for 10 min). After fixation, seedlings were incubated in 200 µl Click-iT reaction cocktail in the dark for 30 min, followed by washing in PBS as above. Root tips were placed onto microscope slides in PBS and imaged using a Leica SP5 confocal laser scanning microscope (Leica, Wetzlar, Germany), with excitation at 488 nm and detection between 516 and 522 nm. Quantification of fluorescence intensity on confocal images was done using ImageJ[64].

**Analysis of *pCYCB1;1:GFP-CYCB1;1* marker lines for cell cycle progression.** The *pCYCB1;1:GFP-CYCB1;1* marker line (in Col-0 background)[79] was used as the male parent in crosses with *dph1-1* and *dph1-2*. F1, F2 and their homozygous F3 progeny were identified based on genotyping by PCR, GFP fluorescence and kanamycin resistance. Homozygous F3 seedlings of wild-type *pCYCB1;1:GFP-CYCB1;1*, *dph1-1 pCYCB1;1:GFP-CYCB1;1*, and *dph1-2 pCYCB1;1:GFP-CYCB1;1* were grown in 0.5× MS medium for 8 days. Seedlings were then stained with 10 mg ml$^{-1}$ propidium iodide (Sigma) for 1 min, rinsed briefly, and mounted on a microscopic slide in ultrapure water. For confocal laser scanning microscopy (Leica SP5), EGFP and propidium iodide were both excited at 488 nm, with detection between 504 and 530 nm for EGFP, and between 605 and 636 nm for propidium iodide. The number of mitotic cells per root tip was quantified by manually counting the cells exhibiting GFP fluorescence[80].

**MDC staining.** MDC (monodansylcadaverine, Sigma) staining was conducted as described[38]. In brief, 1-week-old wild-type and *dph1* mutant seedlings cultivated in 0.5× MS medium were immersed in 0.05 mM MDC in PBS (phosphate buffered saline, 137 mM NaCl, 2.7 mM KCl, 8 mM Na$_2$HPO$_4$, and 2 mM KH$_2$PO$_4$, pH 7.4) for 10 min, followed by two washes in PBS. Roots were imaged using a fluorescent microscope (Imager.M2, Zeiss) with a DAPI filter or under bright field.

**Detection of GFP-ATG8a punctae**. A transgenic line expressing the GFP-ATG8a chimeric fusion protein in the Col-0 background[81] was used as the male parent in a cross with *dph1-1*. F1, F2 and their homozygous F3 progeny were identified based on genotyping by PCR, GFP florescence, kanamycin and hygromycin resistance. Homozygous F3 seedlings of wild-type GFP-ATG8a and *dph1-1* GFP-ATG8a were grown in 0.5× MS medium for 7 days. Ten to twelve seedlings were transferred into one well of a polystyrene six-well plate containing 2 ml liquid 0.5× MS salts with vitamins and 0.5% (w/v) sucrose. AZD-8055, or an equivalent volume of the solvent DMSO, was added at a final concentration of 2 μM. Twenty-four hours later, GFP fluorescence was imaged in root tips using confocal laser scanning microscopy (Leica SP5), followed by manual counting of *punctae*.

**Multielement analysis of shoot tissues**. Freshly harvested rosette tissues were washed twice in ddH$_2$O and flash-frozen in liquid nitrogen, then ground to a powder. The powder was freeze-dried (Alpha 1–4 LDplus, Martin Christ, Osterode, DE) and then equilibrated in ambient air for 2 days. Subsamples of 7–28 mg of dried leaf powder were weighed into PFA microwave vessels (CEM GmbH, Kamp-Lintfort, DE), manually suspended in 3 ml 65% (w/w) HNO$_3$ (analytical grade, Sigma–Aldrich, Steinheim, DE), digested with microwave assistance using temperature ramping to 190 °C within 25 min and holding for 15 min (MarsXpress, CEM GmbH, Kamp-Lintfort, DE), transferred into 15 ml polypropylene screw-cap tubes (Sarstedt AG & Co, Nümbrecht, Germany) and filled up to a total of 10 ml with ultrapure water (Milli-Q, Merck Millipore, Darmstadt, Germany). Multi-element analysis of the sample solution was performed using Inductively Coupled Plasma Atomic Emission Spectrometry (iCAP 6500 Duo, ThermoFisher Scientific, Dreieich, DE), following calibration with a blank and a series of five multielement standards that were manually pipetted from single-element standard solutions for 17 elements commonly detected in Arabidopsis (AAS Standards; Bernd Kraft, Duisburg, Germany). The precision of measurements was validated by measuring a sample blank and an intermediate calibration standard solution, as well as digests of certified reference material (Virginia tobacco leaves, INCT-PVTL 6; Institute of Nuclear Chemistry and Technology, PL), before and after each set of ~50 samples.

**Reporting summary**. Further information on research design is available in the Nature Research Reporting Summary linked to this article.

## Data availability

RNA-seq data are available from ArrayExpress (www.ebi.ac.uk/arrayexpress, E-MTAB-11206), scripts from https://github.com/hzhang1990/RNA-seq. Mass spectrometry proteomics data have been deposited to the ProteomeXchange Consortium via the PRIDE[82] partner repository with the dataset identifier PXD034873 [http://www.ebi.ac.uk/pride]. Source data are provided with this paper.

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

## Acknowledgements

We thank J. D. Dinman (University of Maryland, MD, USA), M. Bendahmane (INRA CNRS ENS Université de Lyon, Lyon, France), C. Grefen (Ruhr University Bochum, DE), S. Üstün (Ruhr University Bochum), D. Leister (LMU Munich, DE), and Arp Schnittger (University of Hamburg, DE) for material, S. Üstün for advice, and M. Leibinger, D. Fischer, J. Herting, T. Stützel (Ruhr University Bochum) for access to equipment. In our group, we thank P. Düchting for multielement analysis, and A. Aufermann, M. Pullack, J. Schab, K. Hagemann, and S. Kleinhubbert for technical assistance.

## Author contributions

H.Z., R.S., and U.K. designed the research. H.Z. performed most experiments, yeast experiments were done by K.Ü. and H.H. J.Q., L.A., K.M., U.B., X.G., L.C., A.S., N.J., and M.P. contributed single experiments or parts of experiments. H.Z., L.A., U.B., M.W., R.S., and U.K. analyzed data. H.Z. and U.K. wrote the manuscript. All authors edited the manuscript.

## Funding

This work was funded by the DFG Research Priority Program SPP1927 "Iron-Sulfur for Life" grants Kr1967/17-1 to U.K., Scha750/21-1 to R.S., AD178/7-1 to L.A., with contributions from DFG Kr1967/3-3, RTG 2341 MiCon, and ERC-AdG LEAP-EXTREME (788380) to U.K, DFG 235736350 WI 3560-2 to M.W., and Consejo Nacional de Ciencia y Tecnología (CONACyT) fellowship no. 448801 to L.C. (U.K.). We acknowledge support by the DFG Open Access Publication Funds of the Ruhr-Universität Bochum. Open Access funding enabled and organized by Projekt DEAL.

## Competing interests

K.M. and U.B. are employed by Roche which has an interest in the diagnosis and treatment of human diseases, and they are co-inventors on patent applications related to diphthamide analyses in oncology. The remaining authors declare no competing interests.
