## [Peer Review File · Nature Communications]

Translational fidelity and growth of *Arabidopsis* require stress-sensitive diphthamide biosynthesisREVIEWER COMMENTS

Reviewer #1 (Remarks to the Author):

Zhang et al. took a reverse genetics approach to determine whether plants, like fungi and animals, modify eEF2 with diphthamide. They identify an orthologue of the gene that encodes the first committed step of diphthamide biosynthesis, DPH1, and then comprehensively characterize growth and developmental defects in *dph1* mutants. Throughout, this report is thorough, experiments are well-executed, and the manuscript is clear and well-written, which made reviewing Zhang et al.'s work a pleasure.

Crucially, although diphthamide modification of eEF2 is well-established in humans and yeast, the functional importance of diphthamide modification remains somewhat controversial, and the evolution of this peculiarly "unique" post-translational modification is under scrutiny, as it becomes clear that this pathway has been lost in some lineages of Archaea. Therefore, I think this is a timely and important contribution, and well-suited for publication in Nature Communications, because it will attract a diverse audience of biologists studying the molecular mechanisms of translation, biomedical consequences of diphthamide-related defects, and plant signaling networks.

That said, I have a few major concerns about sections where the paper veers off from the focused, comprehensive characterization of *dph1* to speculate about downstream impacts of the *dph1* on other cellular pathways, especially relating to autophagy and TOR signaling. Briefly, the evidence presented that autophagy is induced and that TOR activity is attenuated is not convincing, and often points to the opposite conclusion. I urge the authors to reevaluate these sections and consider either removing them, openly discussing alternative and conflicting interpretations of their data, or conduct rigorous additional experiments (with an open mind!) to definitively address these hypotheses.

The effect of *dph1* on autophagy:

(1) I agree that the MDC data presented suggest that autophagy could be induced, but MDC staining is not typically considered sufficient evidence for autophagosome formation (see Klionsky et al., 2008, Autophagy), since this acidotropic dye can label other subcellular compartments.

(2) Autophagy induction is also typically reflected by a transcriptional increase in the expression of autophagy genes, which are otherwise constitutively repressed; in the *dph1* transcriptomes, however, none of the hallmarks of autophagy (e.g., ATG8) are induced.

(3) The only other evidence of autophagy presented is an induction of NBR1 total protein levels. Typically, NBR1 mRNA is induced during autophagy or when TOR activity is attenuated (for evidence in Arabidopsis, see Dong et al., 2015, Frontiers in Plant Science, and Scarpin et al., 2020, eLife). NBR1 total protein levels typically decrease upon autophagy induction, however, because NBR1 is also degraded by autophagy. So, most reports that I've seen show lower levels of NBR1 by Western blot, or perhaps aggregation of NBR1-GFP into puncta with ATG8. In other words, the increase in steady-state NBR1 protein levels without a transcriptional induction of NBR1 might actually suggest that autophagic flux is inhibited in *dph1*.

(4) In a related experiment, Extended Data Fig. 6g, in addition to biological replication, these blots should be experimentally replicated (technical replication, I suppose). The lanes clearly didn't run evenly, and I don't know which regions were used for quantification, but I don't really see the strong effect of *dph1* on protein ubiquitination reported by the authors. A lack of effect might not be surprising, though, since there's only a slight induction of ribosome frameshifting. I'd also note that misfolded proteins might not be soluble, so this assay might not be the best way to detect the changes the authors hope to report.

To address these issues, additional experiments would be required, such as direct demonstration that GFP-ATG8 cleavage is induced in the *dph1* background. Ideally, at least two or three lines of evidence (using established standard techniques) should support induction of autophagy, if it is, indeed, induced at all. The yeast data aren't as immediately relevant, to my mind, although I would note as a minor concern that the mutant and wild-type lanes should be compared in a single Western blot (it looks like the mutants were run separately from WT), and this experiment would need to be replicated (or at least, the replication better described).

The effect of dph1 on TOR activity:

(1) The blots showing S6K-pT449 and total S6K levels need to be shown, ideally not just one, but all of the replicated experiments conducted in the lab. Quantifying changes in S6K-pT449/S6K is notoriously challenging in Arabidopsis, and results of these Westerns can be overinterpreted. Without seeing these, I can't readily evaluate these results.

(2) The "hypersensitivity" of dph1 to AZD-8055 needs to be quantified and more thoroughly investigated, if this point is worth raising at all. Simplistically, if dph1 already has a defect in cell proliferation, and AZD-8055 broadly inhibits growth (cell expansion and proliferation), it is not surprising that dph1 plants would be much smaller than wild-type upon treatment with AZD-8055 (an "additive" effect). It also isn't immediately clear to me that hypersensitivity to AZD-8055 would indicate already-attenuated TOR activity; alternatively, you might expect that if TOR activity is already attenuated in dph1, AZD-8055 might be less disruptive/impactful than for wild-type plants. To put it another way: if mutants and wild-type plants grow similarly under mock conditions, but the mutant grew much smaller after supplying low concentrations of AZD-8055 that don't affect wild-type growth, you could easily argue that the mutant is hypersensitive to AZD-8055. To thoroughly conduct this experiment, growth assays would need to be repeated using a range of AZD-8055 concentrations, a sort of "kinetic" experiment to identify some concentration or condition where AZD-8055 clearly has a qualitatively and quantitatively stronger impact on dph1 than it does on wild-type plants.

(3) The RNA-Seq experiment doesn't really support repression of TOR activity as a major defect in dph1. For example, when TOR is attenuated, under diverse conditions, the same results are observed: autophagy-related genes are transcriptionally induced and ribosomal protein genes are repressed. Instead, the dph1 transcriptomes show, if anything, an induction of r-protein genes and repression (or no effect) on autophagy-related genes.

(4) In plants, it is clear that TOR promotes both cell proliferation and expansion, and has strong effects on developmental timing (e.g., flowering time). As shown here, however, dph1 only impacts proliferation, with little to no impact on expansion. If there were other clear evidence that TOR activity is disrupted, I wouldn't be bothered by this, but given that the other lines of evidence are tenuous (or contradictory), this phenotypic disagreement might be worth reconsidering, too.

(5) TCTP1 (a.k.a. TPT1 in humans) levels are also mentioned as a potential proxy for TOR activity. Indeed, TCTP1 protein levels are sensitive to TOR inhibition across eukaryotes, but probably due to translational regulation, not transcriptional regulation (TCTP1 mRNAs begin with a canonical 5'TOP motif regulated by TOR-LARP1 signaling, see Philippe et al., 2020, PNAS and Scarpin et al., 2020, eLife). I couldn't immediately find an instance in the Arabidopsis literature where TCTP1 transcript levels are sensitive to TOR activity. Moreover, the evidence that TCTP1 acts as a GEF is hotly contested (see Rehmann et al., 2008, FEBS Letters for one example, but there are many) and hasn't been supported outside Drosophila (or, for that matter, even in Drosophila), so I think the discussion of TCTP1 potentially acting upstream of TOR signaling should be removed.

In summary, based on the evidence presented, I'm not yet convinced that TOR activity is disrupted in dph1 mutants, and certainly there is no clear demonstration that TOR signaling mediates any of the defects caused by dph1, since many of the canonical functions of TOR are not disrupted in these mutants. I recommend that the authors critically reevaluate their results, consider alternative hypotheses, and, if the hypothesis that dph1 impacts TOR activity is actually important for this study, they should conduct additional experiments (with an open mind) to fully test this hypothesis.

Lastly, throughout the manuscript, it wasn't always immediately clear to me how many times an experiment was replicated—for example, in Fig. 2b, are the blots shown representative of several replicates, or was the experiment only conducted once? There are several places throughout the manuscript where I would just want to know that the experiment was repeated.

Reviewer #2 (Remarks to the Author):

These research groups lead the functional analysis of the genes of diphthamide modification in yeast and mammalian cells. Zhang et al. extended their study for the physiological roles of eEF2 diphthamide modification in vascular plant, Arabidopsis. By generating loss-of-function AtDPH1 mutants by T-DNA insertion, they confirmed the defect in eEF2 diphthamide modification in dph1 mutant, using MS analysis and the specific antibody which detects diphthamide-unmodified eEF2. They examined the translation fidelity by using mesophyll protoplasts from dph1 mutant with a reporter for programmed -1 frameshifting. As expected, dph1 mutant showed an increased error rate of programmed -1 frameshifting. Homozygous mutant is viable but biomass and primary root length of seedling were about half of wild type. The developmental timing of leaf formation and flowering was unchanged. Leaf palisade cell size was not affected by dph1 mutation but cell number and endopolyploidy decreased. They also found that the ratio of phospho-S6K/total S6K was decreased in dph1 mutant, suggesting the attenuation of TOR activity. Consistent with this observation, TCTP1, a TOR signaling related protein, was nearly halved in dph1 mutant. Transcriptome analysis indicated the upregulation of abiotic stress response in dph1 mutant. Interestingly, they found that heavy metal toxicity by Cu and Cd was correlated with the accumulation of diphthamide-unmodified eEF2.

Specific comments:

In the absence of dph1, dph5 binding to eEF2 could be enhanced, and dph5 may inhibit eEF2 function. Their previous study indicated that the growth phenotype of dph2 mutant yeast under thermal or chemical stress is sensitive to eEF2 down-regulation. I am wondering if dph1 phenotype could be rescued by eEF2 overexpression.

In mice, defects in genes responsible for diphthamide synthesis can cause embryonic lethality with an abnormality in cranial neural crest, neurodevelopment, and digit formation. Growth defects are also reported in mouse embryos. In the reported MS analysis, they found growth defects and attenuation of TOR signaling in dph1 mutant. The decrease in translation in dph1 mutant could be placed as an upstream of TOR, however, they need to show that the translation is indeed attenuated in seedlings.

They showed dph1 mutant seedlings reduced cell proliferation but not cell size. Which steps of cell proliferation are affected in dph1 mutant? Cell cycle analysis can be performed.

They showed the activation of autophagy in dph1 mutant. Programmed -1 frameshifting may cause an increase of misfolded proteins. They can evaluate the accumulation of such proteins in dph1 mutant seedlings.

The reduction of diphthamide-modified eEF2 by Cu or Cd is interesting. Does Cu or Cd reduce ACP intermediate formation in cells (yeast, MCF-7, or mesophyll protoplasts)?

Reviewer #3 (Remarks to the Author):

The article by Zhang et al. provides multiple pieces of evidence of the role of diphthamide modification on eEF2. The experiments are well performed and the article is well structured and easy to read.

This article provides the following evidence:

1. The DPH1 protein (involved in the first committing step of diphthamide synthesis) is conserved in Arabidopsis. The Arabidopsis protein presents a conserved His (H700), which is the one modified by the addition of diphthamide in yeast (H699) and in humans (H715). This gene is ubiquitously expressed in Arabidopsis tissues and the protein is localized, as expected, in the cytoplasm.
2. They identified two dph1 mutants that accumulates unmodified (diphthamide) eEF2. These mutants are hypersensitive to hygromycin compared to the wild type and these mutants show elevated -1 frameshifting error rates.

3. The *dph1* mutants show reduced biomass and reduced primary root length at the seedling stage, reduced area (with no differences in palisade cell size but a reduced number of cells and endoploidy levels) in leaves, and a reduced meristematic zone in the roots (with lower number of cells and an increase in cell length).
4. The *dph1* mutants are hypersensitive to a treatment with a TOR inhibitor and show a reduced TOR activity (based on the decrease in S6K phosphorylation). Consistent with the negative role of TOR in autophagy, these mutants display an enhanced number of autophagosomes and a higher accumulation of the cargo protein NBR1.
5. *dph1* mutants show changes at the transcriptional level in genes related to the response to stress. Specifically, *dph1* mutants show an upregulation of genes involved in abiotic stress.
6. Treatments with heavy metals reduced the amount of modified eEF2 and correlates with decreased growth and decreased biomass.

These results are interesting and provide multiple evidence on the possible role of diphthamide modification on eEF2 in plants. In general terms, the experiments are well performed and support the conclusions.

My main concern is that this study does not deepen in the biological aspect of the findings. This deeper analysis is required to provide additional information on current knowledge of diphthamide and DPH1 already known in other eukaryotes. In this sense, some experiments are similar and provide also similar results to the ones described in yeast and mammals (this is the case of the role of diphthamide modification in translation frameshift fidelity, the hypersensitive phenotype of mutants involved in eEF2 modification and of eEF2 mutants to hygromycin and to other translational inhibitors in yeast (Ortiz et al., 2006), the hypersensitive phenotype to TOR inhibitors and the reduced number of cells in culture of mutants with loss of diphthamide modification which is enhanced in the eEF2 undersupply background (Hawer et al., 2018).

Since some effects have been already described in other systems, this study seems to suggest that the role of diphthamide modification on eEF2 in plants seems quite conserved. This is, without any doubts, interesting but, unfortunately, reduces the novelty of the results.

In my opinion, to increase novelty required for this type of journal, authors should focus in one of the aspects and delve into it, providing further evidence to what is already known in other eukaryotes.

Based on the role of this modification in translation fidelity, it seems crucial to carry out analyses at the level of translation that could allow to identify the direct targets of diphthamide regulation.

RNAseq analysis does not provide this information and probably only reflects indirect targets of the regulation. Authors could characterize in detail the role of DPH1 and diphthamide in translation, analyzing whether this modification affects specifically the translation of specific mRNAs. It has to be taken into account that a high number of translational regulators could have a dual role, in the one hand modifying general translation and in the other fine-tuning the specific translation of subsets of mRNAs. To carry out this analysis in depth, they can carry out a Riboseq analysis with wt and *dph1* mutants. This analysis would help to identify those genes specifically affected by the -1 frameshift error in the absence of diphthamide, providing more light to the role of diphthamide modification in eukaryotes. Furthermore, this analysis could provide additional details on the role of diphthamide in the control of the cell cycle or in the downregulation of TOR activity in the *dph1* mutants.

Alternatively, other translational analyses or even proteomic analyses would be of high interest to understand the role of the cited modification, identifying the targets of this modification.

They also provide evidence that there are a lower number of cells that show a lower level of endoreplication (at least in leaves); however, we do not know how diphthamide regulates the cell cycle and the endoreplication. This is also an interesting question that without a deeper analysis just describes only the surface of the process.

A similar argument could be done for the response to heavy metals. In this case it would be very interesting to know phenotypes of the *dph1* mutants in response to the heavy metals and how translate/proteome is affected (which mRNAs suffer the -1 frameshift during their translation and the effect that this has in the possible generation of a new protein/or in the stability of the proteins) in response to the cited metals.

Other comments:

Line 151-164. Why the size of the cells in the leaf is not altered in the *dph1* mutant (despite the cells

have a reduced ploidy levels), while the meristematic cells in the root have an increased size? Why is it unchanged in the mature cortex? Which is the role of diphthamide in cell cycle and endoreplication progression?

Figures 6 e,f,g. It is hard to see the differences in the accumulation of HSP17.7, HSP17.6 and ubiquitinated proteins. In this case, it would be extremely interesting to show the statistics. Line 239. In general terms the expression of the HSPs is a marker of stress and not specifically of protein aggregation. If these genes are specific markers of protein aggregation, please, include the reference. Line 240. An accumulation of ubiquitinated proteins does not necessarily implies a good clearance of cytosolic protein aggregates. It could be also due to a defect in protein degradation leading to the accumulation of ubiquitinated proteins that could lead to a higher accumulation of aggregates. If dph1 mutants show a higher level of frameshift it is possible that a large portion of the proteins finishes prematurely. Please, provide an explanation to the heat tolerant phenotype of the dph1 mutants.

Lines 249-250. The text establishes that Fig5a and extended Data Fig 7a show immunoblots of the unmodified levels of eEF2, however, none of these panels represent immunoblots.

Extended Figure 7 panel a. Despite the differences in root growth being quite clear in Figure 1, the differences in root growth are not so obvious in extended data Figure 7a. Is this due to the medium or the developmental stage? Please clarify it.

Reviewer #4 (Remarks to the Author):

In the manuscript of Zhang et al., the authors perform a functional characterization of diphthamide modification on eEF2 in Arabidopsis. They first establish that a diphthamide modification pathway is also found in plants, as is the case in other eucaryotes. They identify the genes needed for the histamine modification in the Arabidopsis eEF2 protein, characterize the plant phenotype in loss of function (knock-out) and (gain of function) complementation lines for DPH1 and investigate the functionality of the modification on plant performance under abiotic stress.

The findings are novel, in fact it is the first diphtamide manuscript specifically focusing on the function of this modification in plants that I could find (apologies if I missed some!). Given the conserved mechanism behind the modification, I believe that this will interest a broad readership.

In more detail: Upon identifying the genes that are orthologues to the yeast and human enzymes of this linear reaction in several plant species they 2 T-DNA insertion mutants catalyzing the first step of the diphtamide biosynthesis in Arabidopsis thaliana (At), and have generated multiple complemented lines under the native DPH1 promoter.

The gene characterization is extensive and well rounded, encompassing phenotypic (including intracellular localization), transcript (including RNASeq) and protein analyses (encompassing western blots and mass spectrometry).

After this initial characterization the manuscript expands on a more systemic response, under abiotic stress conditions: heavy metal and heat stress. Plants were assayed under static heavy metal conditions (constitutively grown at the respective metal concentration) and the growth inhibition and the decrease of eEF2 modification was found to correlate to the increase of heavy metal in the medium.

Generally, I found the methods well described, proper controls were included in the experiments, and the statistics used is appropriate in my opinion.

I was particularly asked to focus on the protein analyses and with that in mind:

The mass spectrometry method was easy to follow and I believe it understandable for repetition by others. I appreciated the combined use of standard western blots (Fig 2b) and MS (Fig 2c) as independent confirmations of the post-translational modification (PTM). The MS detection of the diphtamide modification occurs by detection of a m/z difference after MS/MS peptide fragmentation in

the Orbitrap Fusion instrument, which is also the case for other PTMs (hopefully making the wider investigation of this PTM easily applicable for other labs that work on protein modifications). The visual presentation of the mass spectra (Fig 2b and Extended Data Fig. 4a) is clear.

In general, this is a novel and well-rounded manuscript that after minor improvement, should be shared with the scientific community.

Minor comments are listed below.

1. There is a slight omission in the quantification of Fig 3l, 3m in context of lack of biological reproducibility (no error bars in the figure). The authors cite reference 59 (Dong, Y. et al, 2017) who clearly performed reproducible quantification of S6K-p/S6K ratio- so the method is established. I urge the authors to correct this by quantifying a few more samples to increase the robustness of this result. Ideally in this situation I like to see a representative western blot and loading control as already done on other occasions in this work e.g. in 4g (can be added to extended data).

2. The semi quantitative RT-PCR in Extended Data Fig. 2, uses 2 different cycle settings for actin8 (28 cycles) and DPH1 (30 cycles) this is somewhat unusual to me as the target gene and the control are treated differently. Why was this decision made?

RESPONSE TO REVIEWERS

We would like to thank all reviewers for their thoughtful and constructive comments. We apologize for the delay in our re-submission. The pandemic crisis caused an enormous delay in obtaining viable seeds of an Arabidopsis genotype required for crosses with *dph1*. Please find below our responses and the description of changes made during revision based on the reviewers' comments. On our own initiative, we made some modifications to improve the precision or to clarify the text, or shorten by removing redundancies. We also moved most methods used to generate supplemental data to the supplement.

Reviewer #1:

General comment: Zhang et al. took a reverse genetics approach to determine whether plants, like fungi and animals, modify eEF2 with diphthamide. They identify an orthologue of the gene that encodes the first committed step of diphthamide biosynthesis, DPH1, and then comprehensively characterize growth and developmental defects in *dph1* mutants. Throughout, this report is thorough, experiments are well-executed, and the manuscript is clear and well-written, which made reviewing Zhang et al.'s work a pleasure. Crucially, although diphthamide modification of eEF2 is well-established in humans and yeast, the functional importance of diphthamide modification remains somewhat controversial, and the evolution of this peculiarly "unique" post-translational modification is under scrutiny, as it becomes clear that this pathway has been lost in some lineages of Archaea. Therefore, I think this is a timely and important contribution, and well-suited for publication in Nature Communications, because it will attract a diverse audience of biologists studying the molecular mechanisms of translation, biomedical consequences of diphthamide-related defects, and plant signaling networks.

That said, I have a few major concerns about sections where the paper veers off from the focused, comprehensive characterization of *dph1* to speculate about downstream impacts of the *dph1* on other cellular pathways, especially relating to autophagy and TOR signaling. Briefly, the evidence presented that autophagy is induced and that TOR activity is attenuated is not convincing, and often points to the opposite conclusion. I urge the authors to reevaluate these sections and consider either removing them, openly discussing alternative and conflicting interpretations of their data, or conduct rigorous additional experiments (with an open mind!) to definitively address these hypotheses.

Our response: We thank the reviewer for a very knowledgeable assessment, and we have considered the critical comments in our revision. We have conducted additional experiments addressing autophagy and the phosphorylation status of a direct target of phosphorylation by TOR. We are addressing apparent discrepancies listed in the comments below.

1. **Specific comment:** The effect of *dph1* on autophagy: I agree that the MDC data presented suggest that autophagy could be induced, but MDC staining is not typically

considered sufficient evidence for autophagosome formation (see Klionsky et al., 2008, Autophagy), since this acidotropic dye can label other subcellular compartments.

Our response: We agree with the reviewer.

Changes made:

- In the revision, we are additionally including the following results supporting that autophagy is activated in the *dph1* mutants (including the experiments the reviewer suggested):
 - a) We introduced *dph1-1* into a transgenic GFP-ATG8a line (in the Col-0 genetic background). We additionally report in our revised manuscript that compared to GFP-ATG8a in the wild-type background (GFP-ATG8a WT), the number of GFP-labeled autophagosomal structures is significantly higher in GFP-ATG8a *dph1-1* than in the wild type (newly introduced: Extended Data Fig. 10b,d).
 - b) In addition, the GFP:GFP-ATG8a ratios of band intensities of are more than twice as high in GFP-ATG8a *dph1-1* compared to the GFP-ATG8a WT line (newly introduced: Extended Data Fig. 10e).
 - c) We also show that the levels of lipidated ATG8 (ATG8-phosphatidylethanolamine (PE)) are increased in the *dph1* mutants by about 30% compared to the wild type (newly introduced: Extended Data Fig. 10f).
 - d) Finally, we have now focused and strengthened the yeast data (former Extended Data Fig. 6d is now Extended Data Fig. 11). We hope that this now convincingly shows that compared to the wild type, GFP-ATG8 cleavage is also elevated in *dph1* mutants of *Saccharomyces cerevisiae*. In the *dph1* mutant, this effect depends on the kinase ATG1, which has central role in the regulation of autophagy and is under the direct and indirect control of TOR in yeast¹. We are not expanding on this result and its background much in the revised manuscript because there is no scope given the length restrictions.
- We modified the methods section of the manuscript and the corresponding extended data figure legend accordingly.
- We are now including the description of these additional results in the results text (inserted in line 229 of first submission), and introduced a sentence to qualify our results on NBR1 (inserted in line 232 of first submission) (revision lines 254-264).

2. **Specific comment:** Autophagy induction is also typically reflected by a transcriptional increase in the expression of autophagy genes, which are otherwise constitutively repressed; in the *dph1* transcriptomes, however, none of the hallmarks of autophagy (e.g., ATG8) are induced.

Our response: We agree that the expression of autophagy genes is often up-regulated when autophagy is induced. We feel that the apparent discrepancy could be explained by spatio-temporal dynamics and the differing severity of phenotypes between many published studies and our work.

For example, increased autophagic activity can be caused by the decreased activity of TOR, which is a negative regulator of autophagy^{2,3}. We believe that 35S-AtTOR-

RNAi plants physiologically resemble *dph1* mutants in this aspect: Autophagy is enhanced and plants are in a steady-state more than they would be a short time after triggering autophagy by a specific treatment or condition, as often done in other studies. Indeed, in these *AtTOR*-RNAi plants, autophagy was observed under standard cultivation conditions. The only ATGs that were transcriptionally up-regulated in these *AtTOR*-RNAi plants under these conditions, however, were *AtATG9* and *AtATG18a*. Transcript levels of *AtATG9* and *AtATG18a* in the *AtTOR*-RNAi plants were merely increased approximately 1.75-fold compared to the wild type according to RT-qPCR².

Notably, the RNA used in Liu and Bassham (2010)² published study on *AtTOR*-RNAi plants was extracted from the 8-mm-tip region of roots where autophagy was also observed. Different from this published study, RNA-seq analysis for our submission was done using rosette tissues of 4-week-old plants grown in soil, which may contribute to explaining the fact that we did not detect significantly elevated transcript levels for *AtATG9* and *AtATG18a* or other autophagy genes in *dph1* mutants compared to the wild type. Accordingly, the fact that autophagy gene transcript levels are not significantly increased in *dph1* mutants can be consistent with decreased TOR kinase activity. From the literature, our impression is that the upregulation of transcript levels of autophagy-related genes upon autophagy induction is often only transient⁴.

For these reasons, we believe that the direct analysis of well-established autophagy markers is more informative in this respect than the RNA-seq data in our manuscript, which we obtained for a different purpose.

Changes made:

- Added more precision on the age of plants and tissue type used for RNA-seq in line 188. (revision line 203)
- Modified lines 199-204 in results because their contents were a bit misleading and including too much interpretation here. (revision line 216-220)
- We removed the information on the connection between autophagy and TOR from the results section (line 221-222) to mention this only in the discussion. This reflects better our reasoning of why we did this experiment. (revision lines 239-239)
- We modified text and inserted one sentence summarizing this comparison in the discussion (line 310 of first submission and thereafter). (revision lines 353-358)

3. **Specific comment:** The only other evidence of autophagy presented is an induction of NBR1 total protein levels. Typically, NBR1 mRNA is induced during autophagy or when TOR activity is attenuated (for evidence in Arabidopsis, see Dong et al., 2015, Frontiers in Plant Science, and Scarpin et al., 2020, eLife). NBR1 total protein levels typically decrease upon autophagy induction, however, because NBR1 is also degraded by autophagy. So, most reports that I've seen show lower levels of NBR1 by Western blot, or perhaps aggregation of NBR1-GFP into puncta with ATG8. In other words, the increase in steady-state NBR1 protein levels without a

transcriptional induction of NBR1 might actually suggest that autophagic flux is inhibited in *dph1*.

Our response: We fully agree with the reviewer that this has generally been reported. Yet, there are publications suggesting that NBR1 protein levels are not necessarily decreased when autophagic flux is enhanced according to other markers⁵ or even when NBR1-mediated autophagy flux (e.g. aggregophagy) is enhanced⁶, for example. We feel that in *dph1* mutants, a steady-state will have been reached, distinct from the responses to TOR kinase inhibition after 24 h⁷ or 2 h⁸ profiled by the studies cited (see also 2. above). We fully acknowledge that we do not follow up on this discrepancy exhaustively in our present study. We originally did this experiment based on the data published in Dong *et al.* (2017)⁹ – yet we are still unsure about the interpretation of differently sized bands, and the sizes and properties of the bands we observed in our work resemble the results of Jung *et al.* (2020)¹⁰.

Changes made: see 1. and 2. above and 4. below.

4. **Specific comment:** In a related experiment, Extended Data Fig. 6g, in addition to biological replication, these blots should be experimentally replicated (technical replication, I suppose). The lanes clearly didn't run evenly, and I don't know which regions were used for quantification, but I don't really see the strong effect of *dph1* on protein ubiquitination reported by the authors. A lack of effect might not be surprising, though, since there's only a slight induction of ribosome frameshifting. I'd also note that misfolded proteins might not be soluble, so this assay might not be the best way to detect the changes the authors hope to report.

Response: Misfolded proteins with non-native conformations resulting from -1 frameshifting during translation are subject to degradation. The ubiquitin/proteasome system plays an important role in protecting the cell from the negative effects of protein misfolding and aggregation¹¹. Ubiquitylation is a common posttranslational modification associated with misfolded proteins for the degradation by the 26S proteasome. So we detected ubiquitin conjugates as a proxy of the levels of misfolded proteins and an early step in the activation of ubiquitin-proteasome system. Many researchers analyze ubiquitin conjugate levels in total protein extracts (Yang *et al.*, 2016; Üstün *et al.*, 2018)^{12,13}, and given our methodology we expect these to comprise both soluble proteins and a considerable proportion of the so-called "insoluble fraction" (2x Laemmli buffer). We agree with the reviewer that the effect of diphthamide on protein ubiquitination is not very strong, and we attribute this to only moderate levels of misfolded proteins given the moderate rates of -1 frameshifting even in *dph1* mutants (Fig. 2d, Extended Data Fig. 4c-d).

Changes made:

- We are now showing replicate immunoblots as well as independent experiments (Extended Data Fig. 12c-e) and specify the regions used for quantification. The

effect we reported in our first submission is quantitatively small (20 to 30%), but it is reproducible (Extended Data Fig. 12h).

- We clarified in the Results that the effects of *dph1* mutation on small heat shock protein levels and on the levels of ubiquitin conjugates are small, and we rephrased parts of these sentences for accuracy (modifications in lines 238-245 of first submission). (revision lines 265-273)
- We rephrased parts of the corresponding section in the discussion for accuracy (corresponding to lines 305 to 313 of first submission). (revision lines 340-358)

5. **Specific comment:** To address these issues, additional experiments would be required, such as direct demonstration that GFP-ATG8 cleavage is induced in the *dph1* background. Ideally, at least two or three lines of evidence (using established standard techniques) should support induction of autophagy, if it is, indeed, induced at all. The yeast data aren't as immediately relevant, to my mind, although I would note as a minor concern that the mutant and wild-type lanes should be compared in a single Western blot (it looks like the mutants were run separately from WT), and this experiment would need to be replicated (or at least, the replication better described).

Changes made:

- Additional data for Arabidopsis are now included as requested in Extended Data Fig. 10b-f (see comment 1. above).
- Yeast data now included in a different blot (and more focused), as requested, in Extended Data Fig. 11.

6. **Specific comment:** The effect of *dph1* on TOR activity: The blots showing S6K-pT449 and total S6K levels need to be shown, ideally not just one, but all of the replicated experiments conducted in the lab. Quantifying changes in S6K-pT449/S6K is notoriously challenging in Arabidopsis, and results of these Westerns can be overinterpreted. Without seeing these, I can't readily evaluate these results.

Our response: We repeated this experiment multiple times and changed the source of antibodies in the course of this process for cross-validation and a robust detection of the target proteins. The steady-state levels of total S6K are similar (Fig. 3l; Extended Data Fig. 9b) or notably higher (Extended Data Fig. 9a,c) in *dph1* relative to WT, while the S6K-P levels are decreased in the mutant lines. S6K-P is hardly detectable in seedlings of WT and *dph1* treated with AZD-8055, as expected (Extended Data Fig. 9b,c).

Changes made:

- We are now showing the immunoblots and respective quantification from four independent experiments (Fig. 3l,m; Extended Data Fig. 9).
- We adjusted the results text to the numbers as shown in Fig. 3m (line 182). (revision line 199)

7. **Specific comment:** The “hypersensitivity” of *dph1* to AZD-8055 needs to be quantified and more thoroughly investigated, if this point is worth raising at all. Simplistically, if *dph1* already has a defect in cell proliferation, and AZD-8055 broadly inhibits growth (cell expansion and proliferation), it is not surprising that *dph1* plants would be much smaller than wild-type upon treatment with AZD-8055 (an “additive” effect). It also isn’t immediately clear to me that hypersensitivity to AZD-8055 would indicate already-attenuated TOR activity; alternatively, you might expect that if TOR activity is already attenuated in *dph1*, AZD-8055 might be less disruptive/impactful than for wild-type plants. To put it another way: if mutants and wild-type plants grow similarly under mock conditions, but the mutant grew much smaller after supplying low concentrations of AZD-8055 that don’t affect wild-type growth, you could easily argue that the mutant is hypersensitive to AZD-8055. To thoroughly conduct this experiment, growth assays would need to be repeated using a range of AZD-8055 concentrations, a sort of “kinetic” experiment to identify some concentration or condition where AZD-8055 clearly has a qualitatively and quantitatively stronger impact on *dph1* than it does on wild-type plants.

Our response: In yeast cells, the loss of diphthamide sensitizes cell growth to TOR kinase inhibition by rapamycin^{14,15}. Unfortunately, we feel that we cannot extract sufficiently correct quantitative data from the pictures of young seedlings as shown Fig. 3k of our first submission. We additionally feel that it is inherently difficult to interpret the results of such sensitivity assays when comparing the wild type with a mutant that shows reduced growth under control conditions. Following the reviewer’s comment, we used conditions and a set-up following a published example, and we feel quite confident with our result, yet we feel that results of such assays may depend on plant growth stage and other specifics of the assay conditions.

Changes made:

- As suggested by the reviewer, we conducted a quantitative experiment over a range of inhibitor concentrations, followed by a statistical data analysis for the comparison of the sensitivities of wild type and *dph1* mutant seedlings to TOR inhibitor AZD-8055 (Extended Data Fig. 8). We now interpret our results to be consistent with AZD-8055 hyposensitivity, as the reviewer suggested, following procedures of a previous study reporting AZD-8055 hyposensitivity in a mutant defective in TOR signaling¹⁶, in particular when considering that differential effects on root growth appear to occur after d 4.
- We removed the original Fig. 3k, and we replaced it by average root elongation rates from day 4 to day 7 of AZD-8055 treatment in the wild type and in *dph1* mutants.
- We adjusted the manuscript text according to the results in line 69-71 of the first submission (revision lines 68-70)
- We adjusted results in lines 171-180, with also some shortening of the text that was required to accommodate additions elsewhere. (revision lines 189-200)

8. **Specific comment:** The RNA-Seq experiment doesn't really support repression of TOR activity as a major defect in *dph1*. For example, when TOR is attenuated, under diverse conditions, the same results are observed: autophagy-related genes are transcriptionally induced and ribosomal protein genes are repressed. Instead, the *dph1* transcriptomes show, if anything, an induction of r-protein genes and repression (or no effect) on autophagy-related genes.

Our response: Thank you for pointing this out. For autophagy genes, we observe no change (and not a repression) in *dph1*. We feel that comparisons of our data to expectations based on previously published results should be made with caution. To our knowledge, the existing datasets are not directly comparable to our dataset for several reasons.

First, most previous RNA-seq studies compared untreated plants with treated plants after a relatively short time of treatment by adding a highly effective inhibitor or activator of TOR activity. By contrast, our data, taken together, indicates an only slight decrease in TOR activity in *dph1* mutants, clearly smaller than the decrease caused by TOR inhibitor treatment or TOR-RNAi, for example. Among other consequences of *DPH1* loss-of-function, our RNA-seq comparison between *dph1* mutants and the wild type would identify the transcriptional consequences of a small and constitutive decrease in TOR activity after a long period of time, i.e. in a steady-state. These are not very well established to our knowledge, and a direct comparison with published data could be misleading.

Second, known output processes of TORC1 signaling are based on direct approaches of manipulating TOR activity, i.e. TOR inhibitor treatments, a reduction in the levels of TOR kinase protein or glucose addition to stimulate TOR activity. Such direct major-effect approaches are likely to affect multiple TORC1 signaling output processes in parallel. However, there could well be cellular physiological changes that affect only a subset of the known TOR kinase output processes (i.e., not all of them in concert, or not all of them to an equal extent). The absence of diphthamide modification of eEF2, as observed in Arabidopsis *dph1* mutant lines, could affect a subset of the total known TORC1 signaling output processes. The possibility in plants of selectively triggering specific subsets of TOR kinase-regulated output processes (and not just an all-or-nothing activation/deactivation) is conceivable and could make sense biologically.

Third, many previous publications address one or a subset of TOR signaling outputs. Our summarized knowledge of all possible TOR output processes is assembled from studies using a variety of experimental set-ups, environmental conditions and developmental stages. One has to be aware of this when generating expectations.

Finally, there could be cell type-specific, leaf development-dependent, plant age-dependent and possibly environment-dependent effects in *dph1* mutants, and such effects could be (partially) masked when doing RNA-seq on bulk leaf tissues at a specific plant age (as we did).

We believe that these points, in combination with the previous results from Liu and Bassham (2010) and our responses to 2. and 3. above, can fully explain that we did not observe autophagy gene transcript levels to be upregulated in our RNA-seq data.

Changes made: See comment 2. above.

9. **Specific comment:** In plants, it is clear that TOR promotes both cell proliferation and expansion, and has strong effects on developmental timing (e.g., flowering time). As shown here, however, dph1 only impacts proliferation, with little to no impact on expansion. If there were other clear evidence that TOR activity is disrupted, I wouldn't be bothered by this, but given that the other lines of evidence are tenuous (or contradictory), this phenotypic disagreement might be worth reconsidering, too.

Response: In the revision, we included convincing S6K-p evidence for altered activity of TOR on S6K. We agree that dph1 exhibits decreased cell proliferation, but we did not globally observe decreased cell sizes in *dph1*, whereas both are known in plants with altered TOR kinase levels or activity. We feel that the situation is somewhat more complex regarding the published effects of alterations in TOR on flowering time. We cannot exclude that we would have observed additional effects in *dph1* mutants under different environmental conditions – for example cultivation in long days. Note in this context that we quantified the TOR activity in 14-day-old seedlings grown in 0.5x MS liquid medium with 0.5% sugar. We did not investigate whether TOR activity is continuously attenuated throughout later plant developmental stages and in different cultivation conditions, or whether this is ubiquitous across all tissues, organs and cell types of the plant. See also comments 2., 3., 6. and 8. above.

Changes made: See 2., 3., and 6. to 8. above.

10. **Specific comment:** TCTP1 (a.k.a. TPT1 in humans) levels are also mentioned as a potential proxy for TOR activity. Indeed, TCTP1 protein levels are sensitive to TOR inhibition across eukaryotes, but probably due to translational regulation, not transcriptional regulation (TCTP1 mRNAs begin with a canonical 5'TOP motif regulated by TOR-LARP1 signaling, see Philippe et al., 2020, PNAS and Scarpin et al., 2020, eLife). I couldn't immediately find an instance in the Arabidopsis literature where TCTP1 transcript levels are sensitive to TOR activity. Moreover, the evidence that TCTP1 acts as a GEF is hotly contested (see Rehmann et al., 2008, FEBS Letters for one example, but there are many) and hasn't been supported outside *Drosophila* (or, for that matter, even in *Drosophila*), so I think the discussion of TCTP1 potentially acting upstream of TOR signaling should be removed.

Our response: We regret the misunderstanding. In our manuscript, we do not address TCTP1 transcript levels as a potential proxy for, or output of, TOR activity. Instead, we did an RNA-seq experiment in order to identify downstream processes defective in *dph1* mutants that might provide indications towards a molecular mechanism underlying *dph1* mutant phenotypes. This identified decreased *TCTP1* transcript levels in the *dph1* mutants. We considered *TCTP1* transcript as a major candidate because of previously published data on its function in *A. thaliana* and its very high expression levels in the wild type. We followed up and confirmed that TCTP1 protein levels are also decreased in *dph1* mutants. This is relevant specifically in Arabidopsis because the previously

reported phenotypic consequences of *TCTP1* knock-down are partly similar to the phenotypes of *dph1* mutants (as mentioned in line 216 of first submission). This was all independent of TOR, and we referred to possible links with TOR only in the discussion. The concerns raised here by the reviewer are not central to our manuscript. Instead, firstly, knockdown of TCTP of *Drosophila* had a negative impact on S6K phosphorylation as a reporter of TOR activity, suggesting that it acts upstream of TOR signaling (Hsu et al., 2007, but see below)¹⁷. This is a possible model we have in mind and explained in the discussion of the first submission. As mentioned by the reviewer, TCTP1 can act downstream of TOR, in addition, but – as the reviewer highlights – the sole importance of this would not fit with our data and was not our point. Secondly, *AtTCTP1* can complement the cell proliferation defect (Hsu et al., 2007; Brioude et al., 2010)^{17,18} of a *Drosophila* TCTP loss-of-function mutant, and *vice versa* (Brioude et al., 2010¹⁸; the reason why we emphasize the *Drosophila* findings in our discussion). We believe that these previously published findings are relevant for our discussion and should not be deleted.

According to the comment from the reviewer, we contemplated removing the sentences on animal TORC1 signaling, but we decided against this because they are relevant as background information for balanced reflections on the possible role of TCTP1 in plants, which we are also citing in the discussion. We agree with the reviewer that the reported GEF activity of *Drosophila* TCTP (Hsu et al., 2007) has been contested (we were only aware of Rehmann *et al.*, 2008¹⁹ and Wang et al., 2008²⁰, both of whom used mammalian proteins). Yet, we had cited Dong et al. (2009)²¹ on human TCTP whose results support the findings of Hsu et al. (2007)¹⁷ and – to our impression – attempt to reconcile previously reported discrepancies. They highlight an issue in the modeling results of Rehmann et al. (2008)¹⁹ and suggest that whether or not TCTP is experimentally identified to have GEF activity and/or lead to enhanced S6K phosphorylation seems to be strongly dependent on the experimental conditions and systems used so that there might be additional important factors influencing these processes, such as possibly 14-3-3 proteins, for example (Le et al., 2016)²². While we feel that a coarse overview of the current understanding of TCTP1 in the animal TORC1 pathway (including uncertainties) is important here, an explicit discrimination between *Drosophila* and human and the precise review of the animal literature would be lengthy and are not decisive for the credibility of the data on *Arabidopsis* that we report in this manuscript.

Changes made:

- We deleted the half sentence in line 210-216 referring to the connection in animals between TOR and TCTP1, and we replaced it by a different sentence that we hope is less misleading concerning our rationale. The second of the following two sentences was modified to include an additional detail that further emphasizes the phenotypic similarity between *dph1* mutant and TCTP1 knockdown. (revision lines 226-233)

- We rearranged and altered the section of the discussion lines 288 to 304 to clarify, add precision, highlight the most important aspects, balance, and be complete. (revision lines 316-339, 348-350)

11. **Specific comment:** In summary, based on the evidence presented, I'm not yet convinced that TOR activity is disrupted in dph1 mutants, and certainly there is no clear demonstration that TOR signaling mediates any of the defects caused by dph1, since many of the canonical functions of TOR are not disrupted in these mutants. I recommend that the authors critically reevaluate their results, consider alternative hypotheses, and, if the hypothesis that dph1 impacts TOR activity is actually important for this study, they should conduct additional experiments (with an open mind) to fully test this hypothesis.

Our response: We hope that with the additional evidence in the revision and our responses, the reviewer is now convinced about the decrease in TOR activity in *dph1* mutants. See our responses above to comments 2., 3., 6. to 10.

12. **Specific comment:** Lastly, throughout the manuscript, it wasn't always immediately clear to me how many times an experiment was replicated—for example, in Fig. 2b, are the blots shown representative of several replicates, or was the experiment only conducted once? There are several places throughout the manuscript where I would just want to know that the experiment was repeated.

Our Response: See Figure legends and contents of Figures, Extended Data Figures, Reporting Summary (Life Sciences Study Design, Replication) of first submission. The blots shown in Fig. 2b are representative of two independent experiments. Additionally, the result shown in Fig. 2b is replicated in Fig. 5b. Mentioning this all specifically for all the data shown would be uncommon and lengthen the manuscript.

Changes made: In addition to what we provided with the first submission, we are now including an Excel file containing original data according to journal instructions. Additional replication was performed for some experiments during revision (see above).

Reviewer #2 (Remarks to the Author):

General comment: By generating loss-of-function AtDPH1 mutants by T-DNA insertion, they confirmed the defect in eEF2 diphthamide modification in *dph1* mutant, using MS analysis and the specific antibody which detects diphthamide-unmodified eEF2. They examined the translation fidelity by using mesophyll protoplasts from *dph1* mutant with a reporter for programmed -1 frameshifting. As expected, *dph1* mutant showed an increased error rate of programmed -1 frameshifting. Homozygous mutant is viable but biomass and primary root length of seedling were about half of wild type. The developmental timing of

leaf formation and flowering was unchanged. Leaf palisade cell size was not affected by *dph1* mutation but cell number and endopolyploidy decreased.

They also found that the ratio of phospho-S6K/total S6K was decreased in *dph1* mutant, suggesting the attenuation of TOR activity. Consistent with this observation, TCTP1, a TOR signaling related protein, was nearly halved in *dph1* mutant. Transcriptome analysis indicated the upregulation of abiotic stress response in *dph1* mutant. Interestingly, they found that heavy metal toxicity by Cu and Cd was correlated with the accumulation of diphthamide-unmodified eEF2.

Our response: We agree. Thank you.

Specific comments:

1. **Specific comment:** In the absence of *dph1*, *dph5* binding to eEF2 could be enhanced, and *dph5* may inhibit eEF2 function. Their previous study indicated that the growth phenotype of *dph2* mutant yeast under thermal or chemical stress is sensitive to eEF2 down-regulation. I am wondering if *dph1* phenotype could be rescued by eEF2 overexpression.

Our response: If we understand correctly, based on previously published results on a yeast *dph2* mutant, the reviewer puts forward the hypothesis that the *dph2* phenotype in yeast arises from the inhibition of eEF2 by DPH5 in the absence of DPH2. The reviewer proposes that we test whether an analogous situation arises in *Arabidopsis dph1* mutants, i.e. that we test whether overexpression of eEF2 can rescue the *dph1* phenotype.

We agree that eEF2 overexpression would address the hypothesis of the reviewer indirectly. A lot of additional experiments would be then also be required.

In our manuscript, we proposed a candidate gene for *DPH5* function in plants (Extended Data Fig. 1a), but it has not been functionally characterized. This comment appears to be based on previous observations related to yeast *dph2*. ScDPH2 unequivocally has a different orthologue in *A. thaliana* (*AtDPH2*) that is not addressed in this manuscript (see Extended Data Fig. 1c). There is evidence that DPH1 is the catalytic subunit and DPH2 has other functions, and based on amino acid sequence characteristics and phylogenetic reconstructions, this seems to be conserved in *A. thaliana* (see introduction). We feel that addressing this hypothesis could be more promising in the context of *AtDPH2*, and it goes beyond the scope and the goals of this present manuscript. The reviewer should also note that in yeast, DPH5 can ONLY inhibit eEF2 in a *dph2* mutant when *DPH5* is overexpressed from a galactose-inducible plasmid.

Changes made: none.

2. **Specific comment:** In mice, defects in genes responsible for diphthamide synthesis can cause embryonic lethality with an abnormality in cranial neural crest, neurodevelopment, and digit formation. Growth defects are also reported in mouse embryos. In the reported MS analysis, they found growth defects and attenuation of TOR signaling in *dph1* mutant. The decrease in translation in *dph1* mutant could be

placed as an upstream of TOR, however, they need to show that the translation is indeed attenuated in seedlings.

Our response: We agree that our data suggest that DPH1 acts upstream of TOR. Since DPH1 function modulates a protein of the translational machinery, this machinery acts upstream of TOR. However, it is not yet clear which DPH1-dependent aspect of the translational machinery affects TOR activity, and we are sorry that our manuscript text appears to have suggested otherwise. The decrease in TOR activity in Arabidopsis *dph1* mutants can have a variety of causes, which are not mutually exclusive, including increased -1 frameshifting, which we demonstrate to occur (Fig. 2d), decreased levels of TCTP1 (Fig. 4), enhanced levels of protein aggregates (Extended Data Fig. 12), a possible cellular process that monitors the ribosome as such, or – as the reviewer suggests – a possible decrease in translation. Out of these, only two have so far been implicated in causing a decrease in TOR activity in plants: the accumulation of unfolded or non-folded proteins and – only circumstantially – decreased TCTP1 levels. A decrease in protein biosynthesis is not known to result in lower TOR activity in plants. Instead, lowered TOR activity is known to result in an attenuation of protein biosynthesis of plants.

Changes made:

- We rephrased the sentence in lines 281-284 of the discussion to avoid a misunderstanding. (revision lines 309 to 315)
- We moved part of the discussion up to after line 304 from below (lines 305-310), and we re-arranged and expanded this part of the discussion to clarify the potential complexity of events underlying decreased TOR activity in *dph1*. (revision lines 340-345)
- We quantified global translation rates in five-week-old plants based on the incorporation of isotope-labeled amino acids into proteins. At this developmental stage, overall translation rate in *dph1* mutants was indistinguishable from that in the wild type and the complemented line (Extended Data Fig. 5). In mice, it was also proposed that the lack of diphthamide affects the translation of specific proteins rather than global translation rates²³. We also feel that this is implied in recent work on yeast¹⁵. We added this new result in the results section (line 145). (revision line 145)

3. **Specific comment:** They showed *dph1* mutant seedlings reduced cell proliferation but not cell size. Which steps of cell proliferation are affected in *dph1* mutant? Cell cycle analysis can be performed.

Our response: We now show additional data for cell cycle markers, suggesting that there is a reduced number of proliferating cells in S phase and G2/M phase. Our data are consistent with a delay in leaving the G1 phase in mitotic cells, which was previously reported as a result of *TCTP1* knockdown.

Changes made:

- We crossed the cell cycle reporter line *pCYCB1;1:GFP-CYCB1;1* with *dph1* mutants and subsequently obtained homozygous progeny for experiments. The expression of the mitotic cyclin *CYCB1;1*, which marks G2/M-phase of the cell cycle, is lower in the homozygous crossed lines *dph1 pCYCB1;1:GFP-CYCB1;1* than in wild-type *pCYCB1;1:GFP-CYCB1;1* (Extended Data Fig. 7b,d). Similarly, root meristem activity, as assessed using EdU staining, a reporter of proliferating cells in S phase, is reduced (Extended Data Fig. 7a,c).
 - These results are now described in new text inserted in line 167 of first submission and are very similar to what Li et al. (2017)²⁴ reported in root tips in “Differential TOR activation and cell proliferation in Arabidopsis root and shoot apices”. (revision lines 175-185)
 - We are briefly referring to these new results in the discussion (see reviewer #1, comment 10). (revision lines 316-318)
4. **Specific comment:** They showed the activation of autophagy in *dph1* mutant. Programmed -1 frameshifting may cause an increase of misfolded proteins. They can evaluate the accumulation of such proteins in *dph1* mutant seedlings.
Our response: In the first submission, we observed higher levels of HSP17.6, HSP17.7 and ubiquitin conjugates in *dph1*, which are considered markers of misfolded protein accumulation in Arabidopsis (Extended Data Fig. 12a-h).
Changes made: See also reviewer #1, comments 1, 4 and 5.
5. **Specific comment:** The reduction of diphthamide-modified eEF2 by Cu or Cd is interesting. Does Cu or Cd reduce ACP intermediate formation in cells (yeast, MCF-7, or mesophyll protoplasts)?
Our response: We agree that the observation of heavy metal mediated modulation of ACP intermediate formation and hence diphthamide content is very interesting. Given that the antibodies detect unmodified eEF2 in Cd/Cu-treated Arabidopsis, we can exclude ACP intermediate formation in this system.
Changes made: We found that, similar to Arabidopsis, Cu also reduces the diphthamide-modified eEF2 in MCF-7 cells (Fig. 5i), but not in wild-type yeast cells (Extended Data Fig. 13d). We have described this in the results section (line 265, discussion line 335). It is possible that exposure to Cu²⁺ limits diphthamide content in Arabidopsis and MCF-7 by direct (DPH1-DPH2 FeS cluster modification) or indirect (redox component provision/recycling) interference with diphthamide synthesis. Our inability to observe effects of CuSO₄ on ACP intermediate/diphthamide modification on eEF2 at the tested concentrations in yeast does not imply resistance of the yeast pathway. Instead, we rather believe that it is a consequence of differences in metal uptake or detoxification properties between biological systems. The Cu treatment in Extended Data Fig. 13d reduced yeast growth. (revision lines 293-295, 370-371, 379)

Reviewer #3 (Remarks to the Author):

General comment: The article by Zhang et al. provides multiple pieces of evidence of the role of diphthamide modification on eEF2. The experiments are well performed and the article is well structured and easy to read.

This article provides the following evidence:

- a. The DPH1 protein (involved in the first committing step of diphthamide synthesis) is conserved in Arabidopsis. The Arabidopsis protein presents a conserved His (H700), which is the one modified by the addition of diphthamide in yeast (H699) and in humans (H715). This gene is ubiquitously expressed in Arabidopsis tissues and the protein is localized, as expected, in the cytoplasm.
- b. They identified two *dph1* mutants that accumulate unmodified (diphthamide) eEF2. These mutants are hypersensitive to hygromycin compared to the wild type and these mutants show elevated -1 frameshifting error rates.
- c. The *dph1* mutants show reduced biomass and reduced primary root length at the seedling stage, reduced area (with no differences in palisade cell size but a reduced number of cells and endoploidy levels) in leaves, and a reduced meristematic zone in the roots (with lower number of cells and an increase in cell length).
- d. The *dph1* mutants are hypersensitive to a treatment with a TOR inhibitor and show a reduced TOR activity (based on the decrease in S6K phosphorylation). Consistent with the negative role of TOR in autophagy, these mutants display an enhanced number of autophagosomes and a higher accumulation of the cargo protein NBR1.
- e. *dph1* mutants show changes at the transcriptional level in genes related to the response to stress. Specifically, *dph1* mutants show an upregulation of genes involved in abiotic stress.
- f. Treatments with heavy metals reduced the amount of modified eEF2 and correlates with decreased growth and decreased biomass.
- g. These results are interesting and provide multiple evidence on the possible role of diphthamide modification on eEF2 in plants. In general terms, the experiments are well performed and support the conclusions.

Our response: We appreciate the reviewer's nice summary of our major finding reported in the submitted manuscript.

1. **Specific comment:** My main concern is that this study does not deepen in the biological aspect of the findings. This deeper analysis is required to provide additional information on current knowledge of diphthamide and DPH1 already known in other eukaryotes. In this sense, some experiments are similar and provide also similar results to the ones described in yeast and mammals (this is the case of the role of diphthamide modification in translation frameshift fidelity, the hypersensitive phenotype of mutants involved in eEF2 modification and of eEF2 mutants to hygromycin and to other translational inhibitors in yeast (Ortiz et al., 2006), the hypersensitive phenotype to TOR inhibitors and the reduced number of cells in culture of mutants with loss of diphthamide

modification which is enhanced in the eEF2 undersupply background (Hawer et al., 2018).

Our response: We agree some of the findings reported in the manuscript are similar to what had been reported in yeast and mammals, suggesting the conserved function of diphthamide among different species. However, we report the following results that are novel, important and innovative about the putative roles of diphthamide in the growth and development of Arabidopsis.

1. It was not known before that the diphthamide modification exists in plants.
2. Given strong defects in mammalian mutants and a lack of defects in yeast mutants lacking DPH1 alone, the phenotypic consequence of lack of DPH1 in plants were not predictable based on previous knowledge, and we identified these consequences in detail in our manuscript.
3. We showed the smaller size of *dph1* is a result of reduced cell number in Arabidopsis. Further analysis in our revision has now shown that this can be attributed to decreased cell proliferation and fewer cells in S-phase and G2/M- phase of the cell cycle in *dph1* mutants (see reviewer #2, comment 3).
4. We demonstrated attenuated TOR activity of Arabidopsis *dph1* mutants and activated autophagic activity (the latter additionally in yeast). In the revision, we also address the complexity in TOR inhibitor sensitivity in plants (see reviewer #1, comment 7).
5. Although -1 frameshifting errors were elevated in *dph1* mutants, similar with yeast and human *dph* mutants, we report in the revised manuscript that there is no major effect on overall translation rate.
6. Very importantly, we newly identified the inhibitory effect of Cu and Cd on diphthamide synthesis, linking diphthamide with environmental stresses and demonstrating that a lack of diphthamide on eEF2 is physiologically relevant and not merely the consequence of a mutation generated in the laboratory.
7. We newly identify *TCTP1* expression as a candidate acting downstream of diphthamide in Arabidopsis and highlight in detail the phenotypic similarities between *TCTP1*-RNAi lines and *dph1* mutants.
8. Our RNA-seq results are unique.

Changes made: See reviewer #2, comment 3. Important novel findings (3.) are now more strongly supported by additional data (reviewer #1, 1., 4., 5., 6., 7.). We extended novel finding (6.) to human (reviewer #2, comment 5.).

2. **Specific comment:** Since some effects have been already described in other systems, this study seems to suggest that the role of diphthamide modification on eEF2 in plants seems quite conserved. This is, without any doubts, interesting but, unfortunately, reduces the novelty of the results. In my opinion, to increase novelty required for this type of journal, authors should focus in one of the aspects and delve into it, providing further evidence to what is already known in other eukaryotes.

Our response: This suggestion may seem straightforward at first sight, but based on yeast and mammals it was not clear at all which phenotypes to expect in Arabidopsis *dph1* mutants. Once we showed a growth impairment in Arabidopsis, it was necessary to provide the details of it and to address how it arises. If we had left out the mass spectrometry demonstration of the presence and location of the diphthamide modification on Arabidopsis eEF2 and its (degree of) absence in the *dph1* mutant, reviewers would have asked us to provide this evidence. If we had left out the -1 frameshifting results, reviewers would have demanded that we test this and include them. In Arabidopsis, nobody has ever used a frameshifting reporter – thus this is a methodological innovation in this system. See also reviewer #3, comment 1. In summary, we believe that the results in our manuscript are novel and important.

3. **Specific comment:** Based on the role of this modification in translation fidelity, it seems crucial to carry out analyses at the level of translation that could to identify the direct targets of diphthamide regulation. RNAseq analysis does not provide this information and probably only reflects indirect targets of the regulation. Authors could characterize in detail the role of DPH1 and diphthamide in translation, analyzing whether this modification affects specifically the translation of specific mRNAs.

Our response: This is a very interesting direction, and we have initiated this work, which is complex and a long-term project. In contrast to yeast, it is not straightforward to predict target mRNAs of -1 frameshifting in plants, because the prone endogenous sequences and involved mechanisms have not been identified in this biological system. Both the relevant sequence features and in most cases also the functions of the proteins that were very recently proposed (and partly shown) to undergo diphthamide-dependent -1 frameshifting (Zhang et al., 2021)¹⁵ are extremely poorly conserved in Arabidopsis. The results presented in our manuscript underpin that there are additional yet unidentified molecular mechanisms of diphthamide action that are more highly conserved between kingdoms, and we would thus postulate that these involve protein surfaces rather than sequence motifs in specific mRNAs.

Changes made: We tested for a role of diphthamide in global plant translation rate (see reviewer #2, comment 2.).

4. **Specific comment:** It has to be taken into account that a high number of translational regulators could have a dual role, in the one hand modifying general translation and in the other fine-tuning the specific translation of subsets of mRNAs. To carry out this analysis in depth, they can carry out a Riboseq analysis with wt and dph1 mutants. This analysis would help to identify those genes specifically affected by the -1 frameshift error in the absence of diphthamide, providing more light to the role of diphthamide modification in eukaryotes. Furthermore, this analysis could provide additional details on the role of diphthamide in the control of the cell cycle or in the downregulation of TOR activity in the *dph1* mutants. Alternatively, other translational analyses or even

proteomic analyses would be of high interest to understand the role of the cited modification, identifying the targets of this modification.

Our response: We fully agree. For example, ribo-seq will be highly interesting to further address the role of diphthamide in plants, but important aspects of the necessary methodology and data analysis still require development. This complex work has been initiated (and we have results from a pilot experiment), but its completion is clearly beyond the scope of this manuscript. We believe that the data provided in our revised manuscript, with 5 figures and 17 supplementary figures, are indispensable and reach the limits of what can be addressed in a single publication (see also reviewer #3, 3. above).

5. **Specific comment:** They also provide evidence that there are a lower number of cells that show a lower level of endoreplication (at least in leaves); however, we do not know how diphthamide regulates the cell cycle and the endoreplication. This is also an interesting question that without a deeper analysis just describes only the surface of the process.

Our response: Arabidopsis mutants with reduced cell division can undergo more endocycles to enlarge cell size for compensating the reduction in cell number²⁵⁻²⁷. By contrast, leaves of *dph1* mutants showed evidence of reduced endocycling (Extended Data Fig. 6d), implying that the entry into S phase was reduced or delayed in endocycles, as indicated by EdU staining in the revision (Extended Data Fig. 7a,c). In combination, our results thus suggest that *dph1* is also required for the proper progression through endocycles. The similarities between *dph1*, *TCTP1* knockdown and TOR-defective genotypes of *Arabidopsis thaliana* are addressed in our manuscript, and we suggest that *TCTP1* and TOR could be involved in generating these phenotypic effects of *dph1*. See reviewer #2, comment 3.

Changes made: See reviewer #2, comment 3. In the revision, we emphasized the similarity between published phenotypes of *TCTP1* knockdown lines and *dph1* also concerning endocycles (discussion lines 289-292). (revision lines 324-327)

6. **Specific comment:** A similar argument could be done for the response to heavy metals. In this case it would be very interesting to know phenotypes of the *dph1* mutants in response to the heavy metals and how translome/proteome is affected (which mRNAs suffer the -1 frameshift during their translation and the effect that this has in the possible generation of a new protein/or in the stability of the proteins) in response to the cited metals.

Our response: Thank you very much for your great suggestion of analyzing the translomes also of heavy metal-exposed wild-type plants in relation to *dph1* mutant plants and unexposed wild-type plants. This will be very interesting, but we regret that it is beyond the scope of this present manuscript (see also reviewer #3, comments 3. and 4.).

Changes made: Phenotypes of *dph1* mutants in response to metals are now shown in the manuscript (Extended Data Fig. 15 and 16). There is little scope to discuss this result, which we feel does also not allow much discussion) – so we refer to this in line 371 of the revision).

7. **Specific comment:** Line 151-164. Why the size of the cells in the leaf is not altered in the *dph1* mutant (despite the cells have a reduced ploidy levels), while the meristematic cells in the root have an increased size? Why is it unchanged in the mature cortex? Which is the role of diphthamide in cell cycle and endoreplication progression?

Our response: With reduced ploidy one would generally expect smaller-sized cells. The ploidy level of *dph1* mutants is reduced, but the change is quantitatively moderate overall. The proportion of cells with 16C and 32C in *dph1* mutants is approximately 10% lower than in the WT (Extended Data Fig. 6d). Note that the maximal cell areas are clearly a little higher in WT than in *dph1* (Fig. 3d), and these may correspond to those cells with highest ploidy levels, but overall (across all cells of different ploidies) there is no significant difference in the median cell area between WT and *dph1* mutants.

While cortex cell number in the root meristem is significantly reduced in *dph1* mutants, the length of cortex cells is mildly increased in the root meristem and unchanged in the root differentiation zone when compared with WT. *DPH1* has its highest expression in the shoot apex and meristematic regions of lateral and primary root tips (Extended Data Fig. 2b-i). The decreased cortex cell number in the root meristem of the mutant is possibly a result of a defect in the progression of cell division. In line with this hypothesis, we found that the smaller population of dividing cells in the root meristem in the *dph1* mutants as visualized by the EdU staining and GFP-CYCB1;1 marker (Extended Data Fig. 7), suggesting that mutation in *DPH1* affects cell proliferation by reducing the number of cells in S- and G2/M-phases of the cell cycle in the root tips of *dph1*. We believe that increased cortical cell length in the meristematic region of the roots could be a consequence of delayed or reduced numbers of cell divisions in this region. See also reviewer #3, comment 5.).

The root phenotype of *dph1* we reported is quite similar to that reported by Zhou et al. (2011) for Arabidopsis mutant *rrg* (RETARDED ROOT GROWTH)²⁸. This mutant has a shorter primary root length compared with WT, with reduced meristematic cortex cell number, increased length of meristematic cortex cells and unchanged length of mature cortex cells. The number of root meristem cells in the G2/M phase, revealed by the CYCB1;1-GFP marker is decreased in *rrg* mutant compared with WT. Moreover, *rrg* mutants have less endocycling and a decreased ploidy level. The mechanisms of how exactly the combination of these characteristics arises are presently unknown, but we do not believe that our results are contradictory.

Changes made: We introduced a reference to Zhou et al. (2011) in the discussion in line 289 of the first submission. See also reviewer #3, comment 5. (revision lines 324-325)

8. **Specific comment:** Figures 6 e,f,g. It is hard to see the differences in the accumulation of HSP17.7, HSP17.6 and ubiquitinated proteins. In this case, it would be extremely interesting to show the statistics.
Changes made: We have added replicates and statistical analysis results accordingly (Extended Data Fig. 12f-h). Hsp17.6 and Hsp17.7 levels are significantly higher in *dph1-1* than in the wild type. While their levels are quantitatively about twice as high in *dph1-2* as in the wild type, this increase was not statistically significant (Supplementary Fig. 12f,g) – as can be the case with slight effects. Levels of ubiquitin conjugates are significantly higher in *dph1* than in the wild type (Supplementary Fig. 12h). See also reviewer #1, comment 4., and reviewer #2, comment 4.).
9. **Specific comment:** Line 239. In general terms the expression of the HSPs is a marker of stress and not specifically of protein aggregation. If these genes are specific markers of protein aggregation, please, include the reference.
Our response: We are impressed by the attentive reading. We regret very much that we had removed the corresponding citation in the first submission because of the restriction on the number of citations and the fact that we had already cited a similar follow-up publication by the same first author/research group.
Changes made: Now citing correct reference McLoughlin et al. (2016) in line 239. (revision line 267)
10. **Specific comment:** Line 240. An accumulation of ubiquitinated proteins does not necessarily imply a good clearance of cytosolic protein aggregates. It could be also due to a defect in protein degradation leading to the accumulation of ubiquitinated proteins that could lead to a higher accumulation of aggregates.
Our response: We apologize. We did not intend to imply that there is good clearance. We reasoned as follows: If there is increased frameshifting, we would expect increased levels of aggregation and of ubiquitylated proteins. We tested this experimentally, and we confirmed it. Our results do not include direct evidence of how exactly these characteristics arise. We agree that in principle, our results could also indicate impaired protein degradation. However, this would not be parsimonious in the context of the function of DPH1, as this reviewer states also in comment 11 (below).
Changes made: We rephrased the text in lines 239 to 242. See also reviewer #1, comment 4. (revision lines 265-270)
11. **Specific comment:** If *dph1* mutants show a higher level of frameshift it is possible that a large portion of the proteins finishes prematurely. Please, provide an explanation to the heat tolerant phenotype of the *dph1* mutants.
Our response: In the first submission (discussion, lines 313-315), we proposed as an explanation for increased heat stress tolerance of *dph1* mutants a pre-induction of aspects of the heat stress response (Extended Data Figs. 10, 12). It was reported that class I and class II sHSPs (see Extended Data Fig. 12a-b, f-g) protect protein translation factors during heat stress, together with HSP101. Arabidopsis RNA interference (RNAi)

and overexpression lines of such class I and II sHSPs exhibit reduced and enhanced tolerance to heat stress, respectively²⁹.

We agree that we expect that the biosynthesis of part of the proteins finishes prematurely due to increased levels of -1 frameshifting (Fig. 2d, Extended Data Fig. 4c, d). However, it is unclear whether this would affect “a large proportion of proteins” – the results we present rather suggest that a small or moderate proportion of proteins is slightly affected (see also Extended Data Figs. 5-10 and 12). Misfolded proteins with non-native conformations resulting from -1 frameshifting during translation can form aggregates, and they activate cellular processes counteracting aggregation as well as degradation. For example, the ubiquitin/proteasome system plays an important role in protecting the cell from the negative effects of protein misfolding and aggregation¹¹. Therefore, we analyzed the levels of ubiquitylated proteins (Extended Data Fig. 12c-e), among others. See also reviewer #3, comments 8-10.

Changes made: Added McLoughlin et al. (2016) reference (discussion, lines 313-315). (revision lines 351-358)

12. **Specific comment:** Lines 249-250. The text establishes that Fig5a and extended Data Fig 7a show immunoblots of the unmodified levels of eEF2, however, none of these panels represent immunoblots.

Our response: Thank you for pointing this out.

Changes made: We have corrected this mistake by referring to Fig. 5b,e-f instead.

13. **Specific comment:** Extended Figure 7 panel a. Despite the differences in root growth being quite clear in Figure 1, the differences in root growth are not so obvious in extended data Figure 7a. Is this due to the medium or the developmental stage? Please clarify it.

Our response: The primary roots of the wild type grown under control conditions already reached the bottom of the petri dishes (visible in the corresponding photograph) in Extended Data Fig. 7a (now 13a). This gives the impression that the growth difference is less pronounced than it actually is (see Fig. 5a). The reason why we grew the seedlings to the maximum affordable size here was to obtain enough biomass for the analyses shown here and in Fig. 5b. In addition, we apologize that the information given on seedling ages in the first submission was imprecise.

Beyond this, the reviewer is correct: We consistently find that the growth defect in *dph1* mutants is the most severe in juvenile plants, and the difference in size between *dph1* mutants and the wild type decreases progressively with increasing plant age.

Changes made:

- We checked this and are now giving precise information in the revision: now 12 d (instead of two weeks) in Fig. 1, now 18 d (instead of 16 d) in Extended Data Fig. 13. The figure legends have been adjusted accordingly.
- We consistently observe that the magnitude of the growth phenotype of *dph1* mutants decreases with their age. We had not mentioned this to keep the

manuscript short, but are now including it because we believe that this information is relevant for interpreting our results (line 149). (Revision lines 150 to 154)

- We adjusted intermediate conclusion in results to reflect this in line 170. (Revision lines 162-165)

Reviewer #4 (Remarks to the Author)

General comment: In the manuscript of Zhang et al., the authors perform a functional characterization of diphthamide modification on eEF2 in Arabidopsis. They first establish that a diphthamide modification pathway is also found in plants, as is the case in other eukaryotes. They identify the genes needed for the histamine modification in the Arabidopsis eEF2 protein, characterize the plant phenotype in loss of function (knock-out) and (gain of function) complementation lines for DPH1 and investigate the functionality of the modification on plant performance under abiotic stress.

The findings are novel, in fact it is the first diphthamide manuscript specifically focusing on the function of this modification in plants that I could find (apologies if I missed some!). Given the conserved mechanism behind the modification, I believe that this will interest a broad readership.

In more detail: Upon identifying the genes that are orthologues to the yeast and human enzymes of this linear reaction in several plant species they 2 T-DNA insertion mutants catalyzing the first step of the diphthamide biosynthesis in Arabidopsis thaliana (At), and have generated multiple complemented lines under the native DPH1 promoter.

The gene characterization is extensive and well rounded, encompassing phenotypic (including intracellular localization), transcript (including RNASeq) and protein analyses (encompassing western blots and mass spectrometry).

After this initial characterization the manuscript expands on a more systemic response, under abiotic stress conditions: heavy metal and heat stress. Plants were assayed under static heavy metal conditions (constitutively grown at the respective metal concentration) and the growth inhibition and the decrease of eEF2 modification was found to correlate to the increase of heavy metal in the medium.

Generally, I found the methods well described, proper controls were included in the experiments, and the statistics used is appropriate in my opinion.

I was particularly asked to focus on the protein analyses and with that in mind: The mass spectrometry method was easy to follow and I believe it understandable for repetition by others. I appreciated the combined use of standard western blots (Fig 2b) and MS (Fig 2c) as independent confirmations of the post-translational modification (PTM). The MS detection of the diphthamide modification occurs by detection of a m/z difference after MS/MS peptide fragmentation in the Orbitrap Fusion instrument, which is also the case for other PTMs (hopefully making the wider investigation of this PTM easily applicable for other labs that work on protein modifications). The visual presentation of the mass spectra (Fig 2b and Extended Data Fig. 4a) is clear.

In general, this is a novel and well-rounded manuscript that after minor improvement, should be shared with the scientific community.

Our response: We appreciate the reviewer's positive comments on our manuscript.

1. **Specific comment:** There is a slight omission in the quantification of Fig 3l, 3m in context of lack of biological reproducibility (no error bars in the figure). The authors cite reference 59 (Dong, Y. et al, 2017) who clearly performed reproducible quantification of S6K-p/S6K ratio- so the method is established. I urge the authors to correct this by quantifying a few more samples to increase the robustness of this result. Ideally in this situation I like to see a representative western blot and loading control as already done on other occasions in this work e.g. in 4g (can be added to extended data).

Changes made: We are now showing the results of independent experiments used to quantify S6K-P/S6K, and we are providing a statistical analysis (Fig. 3l,m; Extended Data Fig. 9). We adjusted the text accordingly. See also reviewer #1, comment 6.

2. **Specific comment:** The semi quantitative RT-PCR in Extended Data Fig. 2, uses 2 different cycle settings for actin8 (28 cycles) and DPH1 (30 cycles) this is somewhat unusual to me as the target gene and the control are treated differently. Why was this decision made?

Our response: Different cycles were used because the expression levels of *ACTIN8* and *DPH1* are vastly different. *ACTIN8* is a housekeeping gene with a high expression level that is widely used as a constitutively expressed control gene in RT-PCR. The purpose of showing *ACTIN8* is to demonstrate that equal amounts of cDNA were present in comparison between the different samples/genotypes. For this the number of PCR cycles used is not decisive, as long as the PCR is still within the phase during which the amount of product increases in each cycle. Therefore, we used 28 cycles for *Actin8* amplification. We ran 30 cycles for *DPH1* amplification in order to be able to observe the *DPH1* PCR product on the gel. Given the low expression *DPH1*, the PCR reactions for this template are still in a suitable phase after 30 cycles. Note that the goal in these RT-PCRs was merely to obtain qualitative data. Note also that part of these data are independently confirmed in Extended Data Fig. 2p by RT-qPCR and by RNA-seq (Fig. 4e). Extended Data Fig. 2b-l is a confirmation of most of 2a using a different method.

References

- 1 Diaz-Troya, S., Perez-Perez, M. E., Florencio, F. J. & Crespo, J. L. The role of TOR in autophagy regulation from yeast to plants and mammals. *Autophagy* **4**, 851-865, doi:10.4161/auto.6555 (2008).
- 2 Liu, Y. M. & Bassham, D. C. TOR Is a Negative Regulator of Autophagy in *Arabidopsis thaliana*. *Plos One* **5**, e11883, doi:10.1371/journal.pone.0011883 (2010).

- 3 Pu, Y., Luo, X. & Bassham, D. C. TOR-Dependent and -Independent Pathways Regulate Autophagy in *Arabidopsis thaliana*. *Frontiers in Plant Science* **8**, doi:10.3389/fpls.2017.01204 (2017).
- 4 Rose, T. L., Bonneau, L., Der, C., Marty-Mazars, D. & Marty, F. Starvation-induced expression of autophagy-related genes in *Arabidopsis*. *Biol Cell* **98**, 53-67, doi:10.1042/BC20040516 (2006).
- 5 Hafren, A. *et al.* Turnip Mosaic Virus Counteracts Selective Autophagy of the Viral Silencing Suppressor HCpro. *Plant Physiol* **176**, 649-662, doi:10.1104/pp.17.01198 (2018).
- 6 Thirumalaikumar, V. P. *et al.* Selective autophagy regulates heat stress memory in *Arabidopsis* by NBR1-mediated targeting of HSP90.1 and ROF1. *Autophagy* **17**, 2184-2199, doi:10.1080/15548627.2020.1820778 (2021).
- 7 Dong, P. *et al.* Expression profiling and functional analysis reveals that TOR is a key player in regulating photosynthesis and phytohormone signaling pathways in *Arabidopsis*. *Frontiers in Plant Science* **6**, doi:10.3389/fpls.2015.00677 (2015).
- 8 Scarpin, M. R., Leiboff, S. & Brunkard, J. O. Parallel global profiling of plant TOR dynamics reveals a conserved role for LARP1 in translation. *Elife* **9**, doi:10.7554/eLife.58795 (2020).
- 9 Dong, Y. *et al.* Sulfur availability regulates plant growth via glucose-TOR signaling. *Nat Commun* **8**, 1174, doi:10.1038/s41467-017-01224-w (2017).
- 10 Jung, H. *et al.* *Arabidopsis* cargo receptor NBR1 mediates selective autophagy of defective proteins. *Journal of Experimental Botany* **71**, 73-89, doi:10.1093/jxb/erz404 (2020).
- 11 Goldberg, A. L. Protein degradation and protection against misfolded or damaged proteins. *Nature* **426**, 895-899, doi:10.1038/nature02263 (2003).
- 12 Üstün, S. *et al.* Bacteria Exploit Autophagy for Proteasome Degradation and Enhanced Virulence in Plants. *Plant Cell* **30**, 668-685, doi:10.1105/tpc.17.00815 (2018).
- 13 Yang, B. J. *et al.* *Arabidopsis* PROTEASOME REGULATOR1 is required for auxin-mediated suppression of proteasome activity and regulates auxin signalling. *Nat Commun* **7**, 11388, doi:10.1038/ncomms11388 (2016).
- 14 Hawer, H. *et al.* Importance of diphthamide modified EF2 for translational accuracy and competitive cell growth in yeast. *Plos One* **13**, e0205870, doi:10.1371/journal.pone.0205870 (2018).
- 15 Zhang, Y., Lin, Z., Zhu, J., Wang, M. & Lin, H. Diphthamide promotes TOR signaling by increasing the translation of proteins in the TORC1 pathway. *Proc Natl Acad Sci USA* **118**, doi:10.1073/pnas.2104577118 (2021).
- 16 Belda-Palazon, B. *et al.* A dual function of SnRK2 kinases in the regulation of SnRK1 and plant growth. *Nat Plants* **6**, 1345-1353, doi:10.1038/s41477-020-00778-w (2020).
- 17 Hsu, Y.-C., Chern, J. J., Cai, Y., Liu, M. & Choi, K.-W. *Drosophila* TCTP is essential for growth and proliferation through regulation of dRheb GTPase. *Nature* **445**, 785-788, doi:10.1038/nature05528 (2007).
- 18 Brioudes, F., Thierry, A. M., Chambrier, P., Mollereau, B. & Bendahmane, M. Translationally controlled tumor protein is a conserved mitotic growth integrator in animals and plants. *Proc Natl Acad Sci USA* **107**, 16384-16389, doi:10.1073/pnas.1007926107 (2010).

- 19 Rehmann, H. *et al.* Biochemical characterisation of TCTP questions its function as a guanine nucleotide exchange factor for Rheb. *FEBS Letters* **582**, 3005-3010, doi:10.1016/j.febslet.2008.07.057 (2008).
- 20 Wang, X. *et al.* Re-evaluating the Roles of Proposed Modulators of Mammalian Target of Rapamycin Complex 1 (mTORC1) Signaling. *Journal of Biological Chemistry* **283**, 30482-30492, doi:10.1074/jbc.M803348200 (2008).
- 21 Dong, X., Yang, B., Li, Y., Zhong, C. & Ding, J. Molecular basis of the acceleration of the GDP-GTP exchange of human ras homolog enriched in brain by human translationally controlled tumor protein. *J Biol Chem* **284**, 23754-23764, doi:10.1074/jbc.M109.012823 (2009).
- 22 Le, T. P., Vuong, L. T., Kim, A. R., Hsu, Y. C. & Choi, K. W. 14-3-3 proteins regulate Tctp-Rheb interaction for organ growth in *Drosophila*. *Nat Commun* **7**, 11501, doi:10.1038/ncomms11501 (2016).
- 23 Webb, T. R. *et al.* Diphthamide modification of eEF2 requires a J-domain protein and is essential for normal development. *Journal of Cell Science* **121**, 3140-3145, doi:10.1242/jcs.035550 (2008).
- 24 Li, X. *et al.* Differential TOR activation and cell proliferation in Arabidopsis root and shoot apices. *Proc Natl Acad Sci U S A* **114**, 2765-2770, doi:10.1073/pnas.1618782114 (2017).
- 25 De Veylder, L. *et al.* Functional analysis of cyclin-dependent kinase inhibitors of Arabidopsis. *Plant Cell* **13**, 1653-1668, doi:10.1105/tpc.010087 (2001).
- 26 Dewitte, W. *et al.* Arabidopsis CYCD3 D-type cyclins link cell proliferation and endocycles and are rate-limiting for cytokinin responses. *Proc Natl Acad Sci U S A* **104**, 14537-14542, doi:10.1073/pnas.0704166104 (2007).
- 27 Ramirez-Parra, E. & Gutierrez, C. E2F regulates FASCIATA1, a chromatin assembly gene whose loss switches on the endocycle and activates gene expression by changing the epigenetic status. *Plant Physiol* **144**, 105-120, doi:10.1104/pp.106.094979 (2007).
- 28 Zhou, X. *et al.* The Arabidopsis RETARDED ROOT GROWTH gene encodes a mitochondria-localized protein that is required for cell division in the root meristem. *Plant Physiol* **157**, 1793-1804, doi:10.1104/pp.111.185827 (2011).
- 29 McLoughlin, F. *et al.* Class I and II Small Heat Shock Proteins Together with HSP101 Protect Protein Translation Factors during Heat Stress. *Plant Physiol* **172**, 1221-1236, doi:10.1104/pp.16.00536 (2016).

REVIEWERS' COMMENTS

Reviewer #1 (Remarks to the Author):

Overall, I appreciate the efforts made by the authors, and I remain very impressed with the major points of this manuscript. The comprehensive analysis of dph1 mutants, clear demonstration of the conservation of diphthamide modification in plants, and strong evidence that diphthamide modification of eEF2 is affected by Cu and Cd toxicity is all, to my mind, novel, exciting, worthy, and sufficient for publication in Nature Communications. On the other side, to my mind, the authors continue to stretch credulity with several claims that are not well-supported by the evidence they present on a role for DPH1 upstream of TOR in plants, among some other minor points. With the inclusion of the new S6K-pT449 Western blots, I believe that TOR might sometimes be less active in dph1 mutants (although I note that the results are not very reproducible, as clearly demonstrated by the blots in Extended Data Figure 9—contrast panel a with panels b and c. Would an unbiased reader conclude that repressed TOR activity is a consistent, biologically-important output of dph1 that should be employed to explain several dph1 phenotypes?). Nonetheless, as a phenotype, I think it is reasonable to note that TOR activity is likely lower, without any need for invoking a mechanism.

Personally, I wouldn't talk much (if at all) about TOR in this paper, but that's a matter of opinion/perspective and beyond my scope as a reviewer. Short of that, then, I would ask the authors to carefully reconsider various sentences/claims and to soften them wherever possible. (I like the abstract, for instance: just report that TOR is less active and autophagy is relatively induced, without speculation about these being causal or direct downstream effects of dph1 or TCTP1.) I've highlighted a couple examples from the discussion:

Sentences like ll. 314-5: "Decreased TOR activity must account at least partly for the reduced growth of dph1 mutants"—why make this strong assertion without evidence? Why not state "Decreased TOR activity *could* account...?"

340-341: "The activation of autophagy in dph1 mutants can be attributed to decreased TOR activity and an accumulation of misfolded or aggregated peptides..."—again, why not *could* or *might* be attributed?

348-350: "In summary, our results are consistent with a model in which reduced TCTP1 levels lead to the attenuation of TOR activity and the growth phenotypes observed in Arabidopsis dph1 mutants." Sure, but the results are also consistent with the model that TOR acts upstream of TCTP1. The results are also consistent with the model that TOR activity is mildly disrupted as a secondary effect of reduced growth and consequent physiological defects. I could go on with more models, but I think this makes the point: if the evidence presented can support completely opposite models, I'm not sure why you choose to present this model over the others, or present any model about this at all. In the end, all that is shown is that TCTP1 levels are lower in dph1, and then various mechanistic ideas are proposed without experimental support.

A few other points:

The new autophagy evidence is much more convincing than in the original submission, and I appreciate the efforts the authors made to conduct these experiments. I still wonder about the NBR1 result: in extended data Fig. 10, panel g, we see actin levels remaining constant after TOR inhibition despite massive (maybe even 10-fold) decrease in RbCL levels shown by CBB, and then we see ~2/3 NBR1 levels (in this one blot) after TOR inhibition. So, again, naively one would say that inhibiting TOR and inducing autophagy decreases NBR1 levels, at least relative to actin (if not to RbCL), consistent with past reports that NBR1 is degraded by autophagic flux. Nonetheless, the evidence in the rest of this extended figure is sufficient to convince most readers that autophagy is apparently induced in dph1 mutants, through whatever mechanism.

I remain skeptical about extended data figure 12; the HSP17 blots are not great quality (especially panel a) and the anti-ubiquitin blots don't give a clear conclusion. I tried quantifying panel c, for

instance, and did my best to err on the side of more ubiquitinated proteins in dph1—and I only could get to 1.1-fold differences (not 1.33, which should be visually obvious). After adjusting for loading with the anti-actin control, I couldn't see any quantitative difference at all. As stated last time, I don't think this takes away from the paper. Instead, I think the authors could reasonably present these data, analyze them without prejudice, and present the results (e.g., it looks like maybe there is more ubiquitination and more HSP17 expression in dph1, but the effect is slight and the evidence isn't compelling).

So again, to conclude: I greatly appreciate the improvements to the manuscript, and I think that this is a great paper. Although I disagree with some interpretations, I'm ok with disagreement—but the paper should undergo minor revision to best reflect the evidence that is actually presented, rather than making conclusions and then stretching beyond the experiments to find supportive evidence.

Reviewer #2 (Remarks to the Author):

In the revised paper, the authors have adequately answered to my concerns. The new cell cycle analysis helps us for understanding the mechanisms of growth inhibition in dph1 mutant. The data from additional experiments where the addition of Cu increases unmodified eEF2 even in MCF2 is interesting as it suggests that there is regulation of diphthamide modification across species.

Reviewer #3 (Remarks to the Author):

I acknowledge the effort made by the authors to improve some specific aspects of the article. In this sense, and focusing on my own concerns, I acknowledge that in this new version the authors have provided interesting data related to the role of DPH1 in DNA replication and cell cycle and endoreplication progression. I generally agree with the answers to my questions; nevertheless, I still think that analyzing the transcriptome (through Ribo-seq analyses) of the dph1 mutants in this article is important. DPH1 is involved in the diphthamide modification of a translation factor and affects probably specific translation. Therefore, RNAseq data (the analysis of transcription) only provide indirect information. I do not mean that this information is not valuable, but, in my opinion, this information does not uncover the primary targets involved in the dph1 phenotypes. Most probably, Ribo-seq analysis would directly provide a nice view of the process without the need of looking in Arabidopsis for conserved proteins subjected to diphthamide-dependent -1 frameshift in other organisms.

Reviewer #4 (Remarks to the Author):

The authors have met my concerns. Statistics (& replication) has been introduced to the phosphorylation assays adding robustness to their conclusions. My additional question has been answered satisfactory.

Response to reviewers

We thank the reviewers for carefully reading and evaluating our first revision.

Reviewer #1

1. **RC:** Overall, I appreciate the efforts made by the authors, and I remain very impressed with the major points of this manuscript. The comprehensive analysis of dph1 mutants, clear demonstration of the conservation of diphthamide modification in plants, and strong evidence that diphthamide modification of eEF2 is affected by Cu and Cd toxicity is all, to my mind, novel, exciting, worthy, and sufficient for publication in Nature Communications.

Our response: Thank you.

2. **RC:** On the other side, to my mind, the authors continue to stretch credulity with several claims that are not well-supported by the evidence they present on a role for DPH1 upstream of TOR in plants, among some other minor points.
 - (a) With the inclusion of the new S6K-pT449 Western blots, I believe that TOR might sometimes be less active in dph1 mutants (although I note that the results are not very reproducible, as clearly demonstrated by the blots in Extended Data Figure 9—contrast panel a with panels b and c. Would an unbiased reader conclude that repressed TOR activity is a consistent, biologically-important output of dph1 that should be employed to explain several dph1 phenotypes?). Nonetheless, as a phenotype, I think it is reasonable to note that TOR activity is likely lower, without any need for invoking a mechanism.
 - (b) Personally, I wouldn't talk much (if at all) about TOR in this paper, but that's a matter of opinion/perspective and beyond my scope as a reviewer. Short of that, then, I would ask the authors to carefully reconsider various sentences/claims and to soften them wherever possible.
 - i. (I like the abstract, for instance: just report that TOR is less active and autophagy is relatively induced, without speculation about these being causal or direct downstream effects of dph1 or TCTP1.) I've highlighted a couple examples from the discussion:
 - ii. Sentences like ll. 314-5: "Decreased TOR activity must account at least partly for the reduced growth of dph1 mutants"—why make this strong assertion without evidence? Why not state "Decreased TOR activity *could* account...?"
 - iii. 340-341: "The activation of autophagy in dph1 mutants can be attributed to decreased TOR activity and an accumulation of misfolded or aggregated peptides..."—again, why not *could* or *might* be attributed?

iv. 348-350: “In summary, our results are consistent with a model in which reduced TCTP1 levels lead to the attenuation of TOR activity and the growth phenotypes observed in Arabidopsis *dph1* mutants.”

Sure, but the results are also consistent with the model that TOR acts upstream of TCTP1. The results are also consistent with the model that TOR activity is mildly disrupted as a secondary effect of reduced growth and consequent physiological defects. I could go on with more models, but I think this makes the point: if the evidence presented can support completely opposite models, I'm not sure why you choose to present this model over the others, or present any model about this at all. In the end, all that is shown is that TCTP1 levels are lower in *dph1*, and then various mechanistic ideas are proposed without experimental support.

Our response and changes made:

(a) Each of the four independent experiments for which we conducted for this assay unanimously indicated that TOR kinase activity is lower in the two *dph1* mutants than in the wild type (Supplementary Figure 9). The results of the four independent experiments are summarized by showing means (alongside the datapoints for each independent experiment) in Fig. 3m, in support of this. Consequently, our result was qualitatively highly reproducible, only with some quantitative variation in the magnitude of the observed difference. We have to stress here that we approached these experiments openly with regard to their outcome, i.e. as unbiased experimentators, and thus we feel that an unbiased reader would also interpret our observations in the same way as we interpreted them.

We followed published designs of these experiments as commonly used in the TOR community. These published designs always employ very young seedlings. We observed the largest effects in *dph1* mutants and in the presence of added sucrose (note that RNA-seq and protein biosynthesis assays were done with older plants in which the reduced-growth phenotype of *dph1* gradually decreases). So what we quantified in each genotype was the degree to which TOR can be activated under this condition. In order not to lengthen the manuscript inappropriately, we did not go into these details in the text (in the literature, authors do not discuss this, either).

Given that TOR is a central regulator, we feel that its reduced activity is very likely a biologically important output of a lack of *DPH1* function – we can find no convincing argument against this possibility. It is also supported by the conservation of this effect in both Arabidopsis and Saccharomyces. Based on known functions of TOR, reduced TOR activity in *dph1* could explain reduced cell division, reduced growth, reduced chlorophyll content and the activation of autophagy in *dph1* mutants. Indeed, short-term treatment with a TOR kinase inhibitor can mimic some

of the *dph1* phenotypes (cell division/autophagy). We concede that we did not demonstrate the causal involvement of TOR upstream of these phenotypes in *dph1* mutants, and so TOR kinase activity could in principle be affected downstream of one or several of these phenotypes. But we are not aware of published examples. However, since none of our observations pointed against a causal involvement of TOR kinase at least in generating part of these *dph1* phenotypes, explicitly excluding a role for TOR would seem to us like the more substantial misrepresentation of our results.

(b) Our specific answers:

- i. Thank you.
 - ii. We changed this sentence as suggested: "Based on the well-established regulatory functions of TOR³⁰, its decreased activity could account at least in part for the reduced growth of *dph1* mutants (Figs. 1 and 3a-j, Extended Data Fig. 6)."
 - iii. We changed this sentence as suggested: "The activation of autophagy in *dph1* mutants (Extended Data Fig. 10) could be attributed to decreased TOR activity⁵⁶, or alternatively or additionally to an accumulation of misfolded or aggregated peptides in *dph1* mutants, conceivably as a result of translational frameshifting (Figs. 2d and 3l,m, Extended Data Figs. 9, 10, 12)."
 - iv. We are sorry we did not imply our model to exclude alternative models. Our goal was merely to place our results into the context of other published work. We rephrased this section (1st revision lines 343 to 350) to read: "The lack of diphthamide acts indirectly to cause a number of alterations including moderate changes in TOR kinase activity and autophagic flux. We would not expect these changes to necessarily affect all of the previously identified downstream outputs in concert, especially given also the respective differing response and developmental dynamics, environmental conditions and analyzed tissues⁵⁵⁻⁵⁷. Future work should address how our observations on *Arabidopsis dph1* mutants are causally linked, for example by addressing whether reduced TCTP1 levels lead to the attenuation of TOR activity, as the literature may suggest, and whether decreased TOR activity contributes to the cell cycle and growth phenotypes of *dph1* mutants." (now lines 349-357).
3. The new **autophagy evidence** is **much more convincing** than in the original submission, and I appreciate the efforts the authors made to conduct these experiments. I still wonder about the NBR1 result: in extended data Fig. 10, panel g, we see actin levels remaining constant after TOR inhibition despite massive (maybe even 10-fold) decrease in RbcL levels shown by CBB, and then we see ~2/3 NBR1 levels (in

this one blot) after TOR inhibition. So, again, naively one would say that inhibiting TOR and inducing autophagy decreases NBR1 levels, at least relative to actin (if not to RbcL), consistent with past reports that NBR1 is degraded by autophagic flux. Nonetheless, the evidence in the rest of this extended figure is sufficient to convince most readers that autophagy is apparently induced in *dph1* mutants, through whatever mechanism.

Our response and changes made: We agree with the reviewer's suggestion on how to interpret changes in NBR1 protein levels in wild-type plants upon a 24-h TOR inhibitor treatment. We did these treatments during our first revision as a "positive control" to show that we observe the known response mentioned by the reviewer. We believe that in the complete dataset shown in our first revision, other autophagy markers are now far more important than is NBR1, and these other markers are decisive for the interpretation of our data. We outlined in our response letter with the first revision how we view our results on NBR1 protein levels in *dph1* mutants and how we felt it is compatible with the published literature also (Reviewer #1, 3.).

4. I remain skeptical about extended data figure 12; the HSP17 blots are not great quality (especially panel a) and the anti-ubiquitin blots don't give a clear conclusion. I tried quantifying panel c, for instance, and did my best to err on the side of more ubiquitinated proteins in *dph1*—and I only could get to 1.1-fold differences (not 1.33, which should be visually obvious). After adjusting for loading with the anti-actin control, I couldn't see any quantitative difference at all. As stated last time, I don't think this takes away from the paper. Instead, I think the authors could reasonably present these data, analyze them without prejudice, and present the results (e.g., it looks like maybe there is more ubiquitination and more HSP17 expression in *dph1*, but the effect is slight and the evidence isn't compelling). So again, to conclude: I greatly appreciate the improvements to the manuscript, and I think that this is a great paper. Although I disagree with some interpretations, I'm ok with disagreement— but the paper should undergo minor revision to best reflect the evidence that is actually presented, rather than making conclusions and then stretching beyond the experiments to find supportive evidence.

Our response and changes made: Supplementary Figure 12, panels a,b,f,g: We conducted these multiple times, observing quantitatively small but consistent differences. We regret that there are no antibodies of better quality. We clarified in the text of the first revision that effects are quantitatively small but reproducible ("slightly elevated", lines 267 of first revision; previous response letter Reviewer #1, 4., Reviewer #3, 8.). Here we show another experiment (next page):

Supplementary Figure 12, panels c-e, h: In the second revision, we have now re-done the quantification as follows: We did the quantification four times independently, and what is now shown are averaged values. The quantitatively small, but statistically significant differences between *dph1* mutants and the wild type remain. We addressed this difference as a small difference in the first revision (“... were slightly higher”, Line 271; see also previous response letter Reviewer #1, 4). We hope that it becomes clear that we have a balanced view of these results, as well as of the biological importance and relevance of these observations. We present these results because they answer evident and important questions that many readers would ask.

Reviewer #2 (Remarks to the Author):

1. In the revised paper, the authors have adequately answered to my concerns. The new cell cycle analysis helps us for understanding the mechanisms of growth inhibition in *dph1* mutant. The data from additional experiments where the addition of Cu increases unmodified eEF2 even in MCF2 is interesting as it suggests that there is regulation of diphthamide modification across species.

Our response: Thank you for this comment.

Reviewer #3 (Remarks to the Author):

1. I acknowledge the effort made by the authors to improve some specific aspects of the article. In this sense, and focusing on my own concerns, I acknowledge that in this new version the authors have provided interesting data related to the role of DPH1 in DNA replication and cell cycle and endoreplication progression. I generally agree with the answers to my questions; nevertheless, I still think that analyzing the transcriptome (through Ribo-seq analyses) of the *dph1* mutants in this article is important. DPH1 is involved in the diphthamide modification of a translation factor and affects probably specific translation. Therefore, RNAseq data (the analysis of transcription) only provide indirect information. I do not mean that this information is not valuable, but, in my opinion, this information does not uncover the primary targets involved in the *dph1* phenotypes. Most probably, Ribo-seq analysis would directly provide a nice view of the process without the need of looking in Arabidopsis for conserved proteins subjected to diphthamide-dependent -1 frameshift in other organisms.

Our response: Thank you for this remark. We fully agree with the reviewer on the great potential of ribo-seq for eventually providing further important insights. As we mentioned in our response to reviewers accompanying the first revision, we conducted a pilot experiment and concluded from this that substantial developments will still be needed (Reviewer #3, 4.).

Reviewer #4 (Remarks to the Author):

1. The authors have met my concerns. Statistics (& replication) has been introduced to the phosphorylation assays adding robustness to their conclusions. My additional question has been answered satisfactory.

Our response: Thank you for your comment.